# Simultaneous Confidence Intervals for the Ratios of the Means of Zero-Inflated Gamma Distributions and Its Application

**Theerapong Kaewprasert, Sa-Aat Niwitpong ***  **and Suparat Niwitpong**

Department of Applied Statistics, Faculty of Applied Science, King Mongkut's University of Technology North Bangkok, Bangkok 10800, Thailand
* Correspondence: sa-aat.n@sci.kmutnb.ac.th

**Abstract:** Heavy rain in September (the middle of the rainy season in Thailand) can cause unexpected events and natural disasters such as flooding in many areas of the country. Rainfall series that contain both zero and positive values belong to the zero-inflated gamma distribution, which combines the binomial and gamma distributions. Precipitation in various areas of a country can be estimated by using simultaneous confidence intervals (CIs) for the ratios of the means of multiple zero-inflated gamma populations. Herein, we propose six simultaneous CIs constructed using the fiducial generalized CI method, Bayesian and highest posterior density (HPD) interval methods based on the Jeffreys' rule or uniform prior, and method of variance estimates recovery (MOVER). The performances of the proposed simultaneous CI methods were evaluated using a Monte Carlo simulation in terms of the coverage probabilities and expected lengths. The results from a comparative simulation study show that the HPD interval based on the Jeffreys' rule prior performed the best in most cases, while in some situations, the fiducial generalized CI performed well. All of the methods were applied to estimate the simultaneous CIs for the ratios of the means of natural rainfall data from six regions in Thailand.

**Keywords:** zero-inflated gamma distribution; simultaneous confidence intervals; Bayesian estimation; fiducial approach

**MSC:** 62F25

## 1. Introduction

The zero-inflated gamma (ZIG) distribution is suitable for fitting data comprising both non-negative and zero observations: the proportion of zero values is binomially distributed while the positive values follow a gamma distribution with shape and rate parameters. Point and interval estimation and hypothesis testing are the two basic methods used in probability and statistical inference to estimate a model parameter. The CI is the most popular interval estimate method, and numerous researchers have concentrated on the CI for the ZIG distribution. Meanwhile, Kaewprasert et al. [1] broadened the scope by comparing the difference between the means of two ZIG distributions using fiducial method, Bayesian methods, and highest posterior density (HPD). Wang et al. [2] created CIs for the mean of a ZIG distributions based on fiducial inference, parametric bootstrap (PB), and the method of variance estimates recovery (MOVER). Khooriphan et al. [3] proposed Bayesian estimation of rainfall dispersion in Thailand using ZIG distributions. Khooriphan et al. [4] proposed CIs for the ratio of variance of a ZIG distributions using fiducial quantities, Bayesian credible intervals, and HPD intervals. Muralidharan and Kale [5] proposed CIs for the mean of a modified gamma distribution with singularity at zero.

Because of this, the mean is the most widely used unit for measuring central tendency. It is possible to estimate the means from several populations by simultaneously comparing the pairwise differences between their CIs for this parameter provided that

each population is independently and identically distributed (i.i.d.). If we compare two populations using the difference between their means, this difference is probably going to be small, and thus firm and conclusive inference is difficult. Hence, when investigating multiple populations, simultaneously comparing the ratios of the means is more accurate than the differences between the means. Meanwhile, Ren et al. [6] provided simultaneous CIs for the difference between the means of several ZIG distributions based on the fiducial approach. Wang et al. [7] proposed CIs for the difference between the means of two gamma populations. Maneerat et al. [8] constructed Bayesian CIs for a single mean and the difference between two means of delta-lognormal distributions. Maneerat et al. [9] created simultaneous CIs for the difference between the means of several delta-lognormal distributions based on a PB, a fiducial generalized CI (GCI), the MOVER, and Bayesian credible intervals. Malekzadeh and Kharrati-Kopaei [10] constructed simultaneous CIs for the pairwise quantile differences of several heterogeneous two-parameter exponential distributions. Jana and Gautam [11] proposed CIs of difference and ratio of means for zero-adjusted inverse Gaussian distributions using MOVER and Bayesian approaches. Long et al. [12] suggested population mean ratio estimators that used either the first or third quartiles of the auxiliary variable. Indeed, Maneerat and Niwitpong [13] created CIs for the ratio of the means of two delta-lognormal distributions using Bayesian credible intervals, fiducial GCI, and MOVER. Zhang et al. [14] created simultaneous CIs for the ratios of the means of several zero-inflated log-normal distributions using fiducial method and the MOVER. Therefore, datasets of daily rainfall from the six regions in September 2021 were selected. These data comprise positive values that conform to a gamma distribution rather than a lognormal distribution. However, creating simultaneous CIs for the ratios of the means of several ZIG distributions has not yet been reported. Moreover, the applicability of using simultaneous CIs for the ratios of the means of rainfall datasets from several regions that fit ZIG distributions is also an interesting research topic.

In this study, we constructed simultaneous CIs for the ratio of the means of several ZIG populations (k > 2), and we used k = 3 or 6 to estimate the ratio of the means of natural rainfall datasets from six regions in Thailand during September at the height of the rainy season. The fiducial GCI approach, Bayesian, and HPD interval methods based on the Jeffreys'rule or uniform prior, and the MOVER were used to construct simultaneous CIs in this study. The study of Ren et al. [6] served as our inspiration for adopting the fiducial approach to construct simultaneous CIs, while the use of several priors by Maneerat et al. [9] served as our inspiration for developing simultaneous CIs for disparities in the HPD interval and the MOVER. These studies motivated our contribution to this research area of creating simultaneous CIs based on our suggested techniques to clarify the pairwise ratios between the means of multiple ZIG distributions. We calculated the pairwise ratios of the means of daily rainfall records from the Northern, Northeastern, Central, Eastern, Western, and Southern regions of Thailand as a practical demonstration. Importantly, this method could be applied to identify and foretell natural disasters in a specific region

The rest of this paper is organized as follows. In Section 2, we provide the methodologies for the methods to estimate the simultaneous CIs for the ratios of the means of multiple ZIG populations. In Sections 3 and 4, we conduct simulation studies and analyze a rainfall dataset from six regions in Thailand. Finally, a discussion and conclusions are offered in Sections 5 and 6, respectively.

## 2. Materials and Methods

For *k* populations of observations, the probability of observing a zero response is represented $\delta_{i(0)}$ in the *i*th group, while the nonzero observations fit a gamma distribution. For sample $(X_{i1}, X_{i2}, \cdots, X_{in_i})$, $i = 1, 2, \cdots, k$ randomly generated from a ZIG distribution, the $f(x_i)$ is derived as

$$f(x_i) = \begin{cases} \delta_{i(0)} & ; x_i = 0 \\ \delta_{i(1)} g(x_i; \alpha_i, \beta_i) & ; x_i > 0 \end{cases},$$

where $g(x_i; \alpha_i, \beta_i)$ is the probability density function (pdf) of the gamma distribution with shape parameter $\alpha_i$ and rate parameter $\beta_i$, and $\delta_{i(1)} = 1 - \delta_{i(0)}$. The probability of containing zero observations follows binomial distribution denoted as $n_{i(0)} \sim B(n_i, \delta_{i(0)})$, while $n_i = n_{i(0)} + n_{i(1)}$, where $n_{i(0)}$ and $n_{i(1)}$ are the numbers of zero and nonzero values, respectively.

Krishnmoorthy et al. [15] and Krishnmoorthy and Wang [16] showed that $X_i \neq 0$ can be transformed by using the cube-root approximation. As a result, $Y_i = X_i^{1/3} \sim N(\mu_i, \sigma_i^2)$ follows a normal distribution with the mean and variance respectively given by

$$\mu_i = \left(\frac{\alpha_i}{\beta_i}\right)^{1/3} \left(1 - \frac{1}{9\alpha_i}\right) \qquad and \qquad \sigma_i^2 = \frac{1}{9\alpha_i^{1/3}\beta_i^{2/3}}.$$

Since $M_i = \frac{\alpha_i}{\beta_i}$ is the mean of a gamma distribution, $\mu_i$ and $\sigma_i^2$ can be respectively rewritten to yield

$$\mu_i = M_i^{1/3} \left(1 - \frac{1}{9\beta_i M_i}\right) \qquad and \qquad \sigma_i^2 = \frac{1}{9\beta_i M_i^{1/3}}.$$

Thus, $M_i = \left(\frac{\mu_i}{2} + \sqrt{\frac{\mu_i^2}{4} + \sigma_i^2}\right)^3$ is the mean of a gamma distribution and $\lambda_i = \delta_{i(1)}\left(\frac{\mu_i}{2} + \sqrt{\frac{\mu_i^2}{4} + \sigma_i^2}\right)^3$, where $\delta_{i(1)} = 1 - \delta_{i(0)}$, is the mean of a ZIG distribution.

The simultaneous CIs for the ratios of the means of several ZIG populations are what we are interested in creating, and so

$$\lambda_{il} = \lambda_i / \lambda_l = \delta_{i(1)}\left(\frac{\mu_i}{2} + \sqrt{\frac{\mu_i^2}{4} + \sigma_i^2}\right)^3 / \delta_{l(1)}\left(\frac{\mu_l}{2} + \sqrt{\frac{\mu_l^2}{4} + \sigma_l^2}\right)^3,$$

where $i, l = 1, 2, \cdots, k$ and $i \neq l$.

One can respectively replace $\delta_{i(1)}$, $\mu_i$ and $\sigma_i^2$ with their maximum likelihood estimators as follows: $\hat{\delta}_{i(1)} = n_{i(1)}/n_i$, $\hat{\mu}_i = \frac{1}{n_{i(1)}} \sum_{j=1}^{n_{i(1)}} x_{ij}^{1/3}$ and $\hat{\sigma}_i^2 = \frac{1}{n_{i(1)}-1} \sum_{j=1}^{n_{i(1)}} \left(x_{ij}^{1/3} - \hat{\mu}_i\right)^2$. Thus $\hat{\lambda}_i = \hat{\delta}_{i(1)}\left(\frac{\hat{\mu}_i}{2} + \sqrt{\frac{\hat{\mu}_i^2}{4} + \hat{\sigma}_i^2}\right)^3$.

Similarly, the simultaneous CIs for the ratios of the means of several ZIG populations can be defined as

$$\hat{\lambda}_{il} = \hat{\lambda}_i / \hat{\lambda}_l = \hat{\delta}_{i(1)}\left(\frac{\hat{\mu}_i}{2} + \sqrt{\frac{\hat{\mu}_i^2}{4} + \hat{\sigma}_i^2}\right)^3 / \hat{\delta}_{l(1)}\left(\frac{\hat{\mu}_l}{2} + \sqrt{\frac{\hat{\mu}_l^2}{4} + \hat{\sigma}_l^2}\right)^3. \qquad (1)$$

### 2.1. The Fiducial GCI Method

Hannig et al. [17] first introduced the fiducial generalized pivotal quantity (GPQ), a subclass of the GPQ, to construct the simultaneous fiducial approach. Let $X_i = (X_{i1}, X_{i2}, \cdots, X_{in_i})$, $i = 1, 2, \cdots, k$ be a random sample from a ZIG distribution with parameter of interest $\left(\mu_i, \sigma_i^2, \delta_{i(1)}\right)$ across $k$ independent samples and assume that $x_i = (x_{i1}, x_{i2}, \cdots, x_{in_i})$, $i = 1, 2, \cdots, k$ represents $X_i$ observations. The GPQ of $R(X_i; x_i, \mu_i, \sigma_i^2, \delta_{i(1)})$ is referred to as a fiducial GPQ if it satisfies the following two requirements:

1.  The conditional distribution is parameter-free for each $x_i$.
2.  The observed value of $R(X_i; x_i, \mu_i, \sigma_i^2, \delta_{i(1)})$ at $X_i = x_i$, $r(x_i; x_i, \mu_i, \sigma_i^2, \delta_{i(1)})$ is the parameter of interest.

From $Y_{ij} = X_{ij}^{1/3} \sim N(\mu_i, \sigma_i^2)$, $\bar{Y}_i \approx \mu_i + Z\frac{\sigma_i}{\sqrt{n_{i(1)}}}$ and $S_i^2 \approx \sigma_i^2\frac{\chi_{n_{i(1)}-1}^2}{(n_{i(1)}-1)}$ are the sample mean and variance of $Y_{ij}$, respectively, where $Z$ and $\chi_{n_{i(1)}-1}^2$ are standard normal and Chi-squared distributions with $n_{i(1)} - 1$ degrees of freedom, respectively. By replacing $(\bar{Y}_i, S_i)$ with $(\bar{y}_i, s_i)$ and estimating $\mu_i$ and $\sigma_i^2$ from the sample mean and variance, respectively, we obtain

$$\mu_i = \bar{y}_i + \frac{Z}{\sqrt{\chi_{n_{i(1)}-1}^2}}\sqrt{\frac{\left(n_{i(1)}-1\right)s_i^2}{n_{i(1)}}} \qquad and \qquad \sigma_i^2 = \frac{\left(n_{i(1)}-1\right)s_i^2}{\chi_{n_{i(1)}-1}^2}.$$

Accordingly, the respective fiducial GPQs for $\mu_i$, $\sigma_i^2$ and $\delta_{i(1)}$ are

$$R_{\mu_i} = \bar{y}_i + \frac{Z}{\sqrt{\chi_{n_{i(1)}-1}^2}}\sqrt{\frac{\left(n_{i(1)}-1\right)s_i^2}{n_{i(1)}}}, \tag{2}$$

$$R_{\sigma_i^2} = \frac{\left(n_{i(1)}-1\right)s_i^2}{\chi_{n_{i(1)}-1}^2} \tag{3}$$

and

$$R_{\delta_{i(1)}} \sim \frac{1}{2}Beta\left(n_{i(1)}, n_{i(0)}+1\right) + \frac{1}{2}Beta\left(n_{i(1)}+1, n_{i(0)}\right). \tag{4}$$

Subsequently, the fiducial GPQ of $\lambda_i$ is simply

$$R_{\lambda_i} = R_{\delta_{i(1)}}\left(\frac{R_{\mu_i}}{2} + \sqrt{\frac{R_{\mu_i}^2}{4} + R_{\sigma_i^2}}\right)^3.$$

Therefore, the fiducial GPQ for the ratios of the means of several ZIG distributions can be written as

$$R_{\lambda_{il}} = R_{\lambda_i}/R_{\lambda_l} = R_{\delta_{i(1)}}\left(\frac{R_{\mu_i}}{2} + \sqrt{\frac{R_{\mu_i}^2}{4} + R_{\sigma_i^2}}\right)^3 / R_{\delta_{l(1)}}\left(\frac{R_{\mu_l}}{2} + \sqrt{\frac{R_{\mu_l}^2}{4} + R_{\sigma_l^2}}\right)^3. \tag{5}$$

Hence, the $100(1 - \gamma)\%$ two-sided simultaneous CI for $\lambda_{il}$ based on the fiducial GCI method can be written as $L_{il} \leq \lambda_{il} \leq U_{il}$, where $L_{il}$ and $U_{il}$ are the $\gamma/2$th and $(1 - \gamma/2)$th quantiles of $R_{\lambda_{il}}$, respectively.

### 2.2. The Bayesian Methods

The joint likelihood function of $k$ independent ZIG distributions can be obtained from the distribution of $X_i$, for $i = 1, 2, \cdots, k$, with the unknown parameters $\mu_i$, $\sigma_i^2$, and $\delta_{i(1)}$, as follows:

$$L\left(\mu_i, \sigma_i^2, \delta_{i(1)}\right) \propto \prod_{i=1}^{k}\left(1 - \delta_{i(1)}\right)^{n_{i(0)}}\left(\delta_{i(1)}\right)^{n_{i(1)}}(\sigma_i^2)^{-\frac{n_{i(1)}}{2}}$$

$$\times \exp\left[-\frac{1}{2\sigma_i^2}\sum_{j=1}^{n_{i(1)}}\left(x_{ij}^{1/3} - \mu_i\right)^2\right].$$

The Fisher information matrix of the unknown parameters can be represented as the second-order partial derivative of the log-likelihood function with respect to the unknown parameters:

$$
I\left(\mu_i, \sigma_i^2, \delta_{i(1)}\right) = \text{diag}\left[ \begin{array}{cccccc} \frac{n_1}{(1-\delta_{1(1)})\delta_{1(1)}} & \frac{n_1\delta_{1(1)}}{\sigma_1^2} & \frac{n_1\delta_{1(1)}}{2(\sigma_1^2)^2} & \cdots & \cdots & \cdots \\ \frac{n_k}{(1-\delta_{k(1)})\delta_{k(1)}} & \frac{n_k\delta_{k(1)}}{\sigma_k^2} & \frac{n_k\delta_{k(1)}}{2(\sigma_k^2)^2} \end{array} \right].
$$

The Jeffreys' rule and uniform priors used to construct equal-tailed simultaneous CIs and simultaneous HPD intervals are covered in the following subsections.

### 2.2.1. The Jeffreys Rule Prior

The square root of the determinant of the Fisher information matrix is used to calculate the Jeffreys rule prior. It is common knowledge that gamma and binomial distributions comprise a ZIG distribution. From the mean $\lambda_i = \delta_{i(1)}\left(\frac{\mu_i}{2} + \sqrt{\frac{\mu_i^2}{4} + \sigma_i^2}\right)^3$, the parameters of interest are $\mu_i$, $\sigma_i^2$, and $\delta_{i(1)}$; Harvey and Van Der Merwe [18] used the Jeffreys rule prior for these parameters as $p(\sigma_i^2) \propto 1/\sigma_i^3$ and $p(\delta_{i(1)}) \propto (1-\delta_{i(1)})^{-1/2}\delta_{i(1)}^{1/2}$, respectively.

The joint posterior density function can be expressed as the likelihood function and the prior distribution of a ZIG distribution as follows:

$$
p\left(\mu_i, \sigma_i^2, \delta_{i(1)} \mid x_{ij}\right) = \prod_{i=1}^{k} \frac{1}{\text{Beta}\left(n_{i(1)} + \frac{3}{2}, n_{i(0)} + \frac{1}{2}\right)} \left(1 - \delta_{i(1)}\right)^{\left(n_{i(0)}+\frac{1}{2}\right)-1} \delta_{i(1)}^{\left(n_{i(1)}+\frac{3}{2}\right)-1}
$$

$$
\times \frac{\sqrt{n_{i(1)}}}{\sqrt{2\pi\sigma_i^2}} \exp\left(-\frac{n_{i(1)}}{2\sigma_i^2}(\mu_i - \hat{\mu}_i)^2\right) \frac{\left(\frac{(n_{i(1)}+1)\hat{\sigma}_i^2}{2}\right)^{\frac{n_{i(1)}+1}{2}}}{\Gamma\left(\frac{n_{i(1)}+1}{2}\right)}
$$

$$
\times \left(\sigma_i^2\right)^{-\frac{n_{i(1)}+1}{2}-1} \exp\left(-\frac{(n_{i(1)}+1)\hat{\sigma}_i^2}{2\sigma_i^2}\right),
$$

where $\hat{\mu}_i = \frac{1}{n_{i(1)}} \sum_{j=1}^{n_{i(1)}} x_{ij}^{1/3}$ and $\hat{\sigma}_i^2 = \frac{1}{n_{i(1)}-1} \sum_{j=1}^{n_{i(1)}} \left(x_{ij}^{1/3} - \hat{\mu}_i\right)^2$.

The respective posterior distributions of $\mu_i$, $\sigma i^2$, and $\delta_{i(1)}$ are obtained using integration as

$$
p\left(\mu_i \mid x_{ij}\right) \propto \prod_{i=1}^{k} \frac{\sqrt{n_{i(1)}}}{\sqrt{2\pi\sigma_i^2}} \exp\left(-\frac{n_{i(1)}}{2\sigma_i^2}(\mu_i - \hat{\mu}_i)^2\right),
$$

$$
p\left(\sigma_i^2 \mid x_{ij}\right) \propto \prod_{i=1}^{k} \frac{\left(\frac{(n_{i(1)}+1)\hat{\sigma}_i^2}{2}\right)^{\frac{n_{i(1)}+1}{2}}}{\Gamma\left(\frac{n_{i(1)}+1}{2}\right)} \left(\sigma_i^2\right)^{-\frac{n_{i(1)}+1}{2}-1} \exp\left(-\frac{(n_{i(1)}+1)\hat{\sigma}_i^2}{2\sigma i^2}\right),
$$

and

$$
p\left(\delta_{i(1)} \mid x_{ij}\right) \propto \prod_{i=1}^{k} \frac{1}{\text{Beta}\left(n_{i(1)} + \frac{3}{2}, n_{i(0)} + \frac{1}{2}\right)} \left(1 - \delta_{i(1)}\right)^{\left(n_{i(0)}+\frac{1}{2}\right)-1} \delta_{i(1)}^{\left(n_{i(1)}+\frac{3}{2}\right)-1}.
$$

As indicated by $\mu_i(\text{J}) \sim \text{N}\left(\hat{\mu}_i, \frac{\sigma_i^2(\text{J})}{n_{i(1)}}\right)$, $\sigma_i^2(\text{J}) \sim \text{IG}\left(\frac{n_{i(1)}+1}{2}, \frac{(n_{i(1)}+1)\hat{\sigma}_i^2}{2}\right)$, and $\delta_{i(1)}(\text{J}) \sim \text{Beta}\left(n_{i(1)} + \frac{3}{2}, n_{i(0)} + \frac{1}{2}\right)$, respectively, $p(\mu_i \mid x_{ij})$ follows a normal distribution, $p(\sigma_i^2 \mid x_{ij})$

follows an inverse gamma distribution, and $p\left(\delta_{i(1)} \mid x_{ij}\right)$ follows a beta distribution. The result is that $\mu_i(\text{J})$, $\sigma_i^2(\text{J})$, and $\delta_{i(1)}(\text{J})$ can be replaced, resulting in

$$\lambda_{il}(\text{J}) = \delta_{i(1)}(\text{J})\left(\frac{\mu_i(\text{J})}{2} + \sqrt{\frac{\mu_i^2(\text{J})}{4} + \sigma_i^2(\text{J})}\right)^3 / \delta_{l(1)}(\text{J})\left(\frac{\mu_l(\text{J})}{2} + \sqrt{\frac{\mu_l^2(\text{J})}{4} + \sigma_l^2(\text{J})}\right)^3. \quad (6)$$

Therefore, the $100(1 - \gamma)\%$ equal-tailed simultaneous CI and simultaneous HPD intervals for $\lambda_{il}$ based on the Bayesian method are $L_{il}(\text{J}) \leq \lambda_{il}(\text{J}) \leq U_{il}(\text{J})$, where $L_{il}(\text{J})$ and $U_{il}(\text{J})$ are the lower and upper bounds of the intervals, respectively. We computed $L_{il}(\text{HPD.J})$ and $U_{il}(\text{HPD.J})$ using the *HPDinterval* package in the R software package to determine the $100(1 - \gamma)\%$ simultaneous HPD intervals for $\lambda_{il}$.

### 2.2.2. The Uniform Prior

Bolstad and Curran [19] proposed that the uniform priors of $\mu_i$, $\sigma_i^2$ and $\delta_{i(1)}$ are 1 ($p(\mu_i) \propto 1$, $p(\sigma_i^2) \propto 1$ and $p(\delta_{i(1)}) \propto 1$, respectively) because the uniform prior has a constant function for the prior probability. Subsequently, $p(\mu_i, \sigma_i^2, \delta_{i(1)}) \propto 1$ is the uniform prior for a ZIG distribution for which the joint posterior density function is

$$p\left(\mu_i, \sigma_i^2, \delta_{i(1)} \mid x_{ij}\right) = \prod_{i=1}^{k} \frac{1}{\text{Beta}\left(n_{i(1)} + 1, n_{i(0)} + 1\right)} \left(1 - \delta_{i(1)}\right)^{\left(n_{i(0)}+1\right)-1} \delta_{i(1)}^{\left(n_{i(1)}+1\right)-1}$$

$$\times \frac{\sqrt{n_{i(1)}}}{\sqrt{2\pi\sigma_i^2}} \exp\left(-\frac{n_{i(1)}}{2\sigma_i^2}(\mu_i - \hat{\mu}_i)^2\right) \frac{\left(\frac{(n_{i(1)}-2)\hat{\sigma}_i^2}{2}\right)^{\frac{n_{i(1)}-2}{2}}}{\Gamma\left(\frac{n_{i(1)}-2}{2}\right)}$$

$$\times \left(\sigma_i^2\right)^{-\frac{n_{i(1)}-2}{2}-1} \exp\left(-\frac{(n_{i(1)} - 2)\hat{\sigma}_i^2}{2\sigma_i^2}\right),$$

where $\hat{\mu}_i = \frac{1}{n_{i(1)}} \sum_{j=1}^{n_{i(1)}} x_{ij}^{1/3}$ and $\hat{\sigma}_i^2 = \frac{1}{n_{i(1)}-1} \sum_{j=1}^{n_{i(1)}} \left(x_{ij}^{1/3} - \hat{\mu}_i\right)^2$.

The respective posterior distributions of $\mu_i$, $\sigma_i^2$, and $\delta_{i(1)}$ are obtained using integration as

$$p\left(\mu_i \mid x_{ij}\right) \propto \prod_{i=1}^{k} \frac{\sqrt{n_{i(1)}}}{\sqrt{2\pi\sigma_i^2}} \exp\left(-\frac{n_{i(1)}}{2\sigma_i^2}(\mu_i - \hat{\mu}_i)^2\right),$$

$$p\left(\sigma_i^2 \mid x_{ij}\right) \propto \prod_{i=1}^{k} \frac{\left(\frac{(n_{i(1)}-2)\hat{\sigma}_i^2}{2}\right)^{\frac{n_{i(1)}-2}{2}}}{\Gamma\left(\frac{n_{i(1)}-2}{2}\right)} \left(\sigma_i^2\right)^{-\frac{n_{i(1)}-2}{2}-1} \exp\left(-\frac{(n_{i(1)} - 2)\hat{\sigma}_i^2}{2\sigma i^2}\right),$$

and

$$p\left(\delta_{i(1)} \mid x_{ij}\right) \propto \prod_{i=1}^{k} \frac{1}{\text{Beta}\left(n_{i(1)} + 1, n_{i(0)} + 1\right)} \left(1 - \delta_{i(1)}\right)^{\left(n_{i(0)}+1\right)-1} \delta_{i(1)}^{\left(n_{i(1)}+1\right)-1}.$$

Thus, the posterior distributions are $\mu_i(\text{U}) \sim \text{N}\left(\hat{\mu}_i, \frac{\sigma_i^2(\text{U})}{n_{i(1)}}\right)$, $\sigma_i^2(\text{U}) \sim \text{IG}\left(\frac{n_{i(1)}-2}{2}, \frac{(n_{i(1)}-2)\hat{\sigma}_i^2}{2}\right)$, and $\delta_{i(1)}(\text{U}) \sim \text{Beta}\left(n_{i(1)} + 1, n_{i(0)} + 1\right)$, respectively.

To construct the equal-tailed simultaneous CI and simultaneous HPD intervals, $\mu_i(U)$, $\sigma_i^2(U)$ and $\delta_{i(1)}(U)$ can be substituted into Equation (1).

### 2.3. Method of Variance Estimates Recovery (MOVER)

First introduced by Donner and Zou [20], the MOVER approach is applied to construct the $100(1 - \gamma)\%$ two-sided simultaneous CI for $\lambda_{il} = \lambda_i/\lambda_l$, for which $L_{il}(\text{MOVER}) \leq \lambda_{il}(\text{MOVER}) \leq U_{il}(\text{MOVER})$, where $L_{il}(\text{MOVER})$ and $U_{il}(\text{MOVER})$ are the lower and upper bounds of the interval, respectively expressed as

$$L_{il}(\text{MOVER}) = \frac{\hat{\lambda}_i \hat{\lambda}_l - \sqrt{(\hat{\lambda}_i \hat{\lambda}_l)^2 - l_i u_l (2\hat{\lambda}_i - l_i)(2\hat{\lambda}_l - u_l)}}{u_l (2\hat{\lambda}_l - u_l)} \tag{7}$$

and

$$U_{il}(\text{MOVER}) = \frac{\hat{\lambda}_i \hat{\lambda}_l + \sqrt{(\hat{\lambda}_i \hat{\lambda}_l)^2 - u_i l_l (2\hat{\lambda}_i - u_i)(2\hat{\lambda}_l - l_l)}}{l_l (2\hat{\lambda}_l - l_l)}, \tag{8}$$

for $i, l = 1, 2, \cdots, k$ and $i \neq l$.

The parameters of interest in $\lambda_i = \delta_{i(1)} \left( \frac{\mu_i}{2} + \sqrt{\frac{\mu_i^2}{4} + \sigma_i^2} \right)^3$ are $\delta_{i(1)}$, $\mu_i$, and $\sigma_i^2$, for which it is possible to construct CIs. From Hannig's [21] paper on the fiducial GPQ of $\delta_{i(1)}$ in Equation (4), the $100(1 - \gamma)\%$ CI for $\delta_{i(1)}$ can be written as

$$CI_{\delta_{i(1)}} = [l_{\delta_{i(1)}}, u_{\delta_{i(1)}}],$$

where $l_{\delta_{i(1)}}$ and $u_{\delta_{i(1)}}$ are the $(\gamma/2)$-th and $(1 - \gamma/2)$-th quantiles of $\delta_{i(1)}$, respectively.

By using the CI definitions for parameters $\mu_i$ and $\sigma_i^2$ in Equations (2) and (3), respectively, we can define the $100(1 - \gamma)\%$ CI for $\mu_i$ as

$$CI_{\mu_i} = [l_{\mu_i}, u_{\mu_i}],$$

where

$$l_{\mu_i} = \hat{\mu}_i - \frac{Z_{i(\gamma/2)}}{\sqrt{\chi^2_{1-\gamma/2, n_{i(1)}-1}}} \sqrt{\frac{\left(n_{i(1)} - 1\right) \hat{\sigma}_i^2}{n_{i(1)}}},$$

and

$$u_{\mu_i} = \hat{\mu}_i + \frac{Z_{i(\gamma/2)}}{\sqrt{\chi^2_{\gamma/2, n_{i(1)}-1}}} \sqrt{\frac{\left(n_{i(1)} - 1\right) \hat{\sigma}_i^2}{n_{i(1)}}}.$$

Thus, the $100(1 - \gamma)\%$ CI for $\sigma_i^2$ can be written as

$$CI_{\sigma_i^2} = [l_{\sigma_i^2}, u_{\sigma_i^2}],$$

where

$$l_{\sigma_i^2} = \frac{\left(n_{i(1)} - 1\right) \hat{\sigma}_i^2}{\chi^2_{1-\gamma/2, n_{i(1)}-1}},$$

and

$$u_{\sigma_i^2} = \frac{\left(n_{i(1)} - 1\right) \hat{\sigma}_i^2}{\chi^2_{\gamma/2, n_{i(1)}-1}}.$$

By ensuring that $\hat{\mu}_i = \frac{1}{n_{i(1)}} \sum_{j=1}^{n_{i(1)}} x_{ij}^{1/3}$ and $\hat{\sigma}_i^2 = \frac{1}{n_{i(1)}-1} \sum_{j=1}^{n_{i(1)}} \left( x_{ij}^{1/3} - \hat{\mu}_i \right)^2$; $Z_i$, $i = 1, 2, \cdots, k$ follow a standard normal distribution, the $100(1 - \gamma)\%$ MOVER interval for $\lambda_i$ becomes

$$CI_{\lambda_i} = [l_i, u_i].$$

Similarly, we can obtain $CI_{\lambda_l} = [l_l, u_l]$. Therefore, the $100(1 - \gamma)\%$ two-sided simultaneous CI for $\lambda_{il}$ based on the MOVER method can be obtained at $[L_{il}(\text{MOVER}), U_{il}(\text{MOVER})]$, for $i, l = 1, 2, \cdots, k$ and $i \neq l$. This process is specified in Algorithm 1.

---

**Algorithm 1** All six methods.

---

1. Begin loop $M$.
2. Generate $X_i$, $i = 1, 2, \cdots, k$ with sample size $n_1, n_2, \cdots, n_k$ from $\text{ZIG}(\alpha_i, \beta_i, \delta_{i(1)})$.
3. Perform cube-root transformation on $n_{i(1)}$ nonzero observations and estimate $\hat{\delta}_{i(1)}$, $\hat{\mu}_i$, and $\hat{\sigma}_i^2$.
4. Get $\lambda_i$ and $\lambda_l$ by computing the parameter.
   (a) Fiducial GCI: compute $R_{\delta_{i(1)}}$, $R_{\delta_{l(1)}}$, $R_{\mu_i}$, $R_{\mu_l}$, $R_{\sigma_i^2}$ and $R_{\sigma_l^2}$.
   (b) Bayesian and HPD based on Jeffreys rule prior: compute $\delta_{i(1)}(\text{J})$, $\delta_{l(1)}(\text{J})$, $\mu_i(\text{J})$, $\mu_l(\text{J})$, $\sigma_i^2(\text{J})$ and $\sigma_l^2(\text{J})$.
   (c) Bayesian and HPD based on uniform prior: compute $\delta_{i(1)}(\text{U})$, $\delta_{l(1)}(\text{U})$, $\mu_i(\text{U})$, $\mu_l(\text{U})$, $\sigma_i^2(\text{U})$ and $\sigma_l^2(\text{U})$.
   (d) MOVER: compute $l_{\delta_{i(1)}}$, $l_{\delta_{l(1)}}$, $u_{\delta_{i(1)}}$, $u_{\delta_{l(1)}}$, $l_{\mu_i}$, $l_{\mu_l}$, $u_{\mu_i}$, $u_{\mu_l}$, $l_{\sigma_i^2}$, $l_{\sigma_l^2}$, $u_{\sigma_i^2}$ and $u_{\sigma_i^2}$.
5. Repeat steps (3) and (4) a total $m$ ($m = 2000$) times.
6. Compute the $100(1 - \gamma)\%$ simultaneous CI for $\lambda_{il}$.
   (a) Fiducial GCI: compute $R_{\lambda_{il}}(\gamma/2)$ and $R_{\lambda_{il}}(1 - \gamma/2)$ using Equation (5).
   (b) Bayesian based on Jeffreys rule prior: compute $\lambda_{il}(\text{J})(\gamma/2)$ and $\lambda_{il}(\text{J})(1 - \gamma/2)$ using Equation (6).
   (c) HPD based on Jeffreys rule prior: using Equation (6) to compute $\lambda_{il}(\text{HPD.J})$ by utilizing the *HPDinterval* package.
   (d) Bayesian based on uniform prior: compute $\lambda_{il}(\text{U})(\gamma/2)$ and $\lambda_{il}(\text{U})(1 - \gamma/2)$.
   (e) HPD based on uniform prior: compute $\lambda_{il}(\text{HPD.U})(\gamma/2)$ and $\lambda_{il}(\text{HPD.U})(1 - \gamma/2)$.
   (f) MOVER: Compute the simultaneous CIs based on MOVER using Equations (7) and (8).
7. End loop $M$.

---

## 3. Simulation Study

We conducted simulation studies to assess how well the proposed methods perform with finite samples using the following requirements:

1. Coverage probability (CP): the percentage of times that the true parameter value is contained within the interval.
2. Expected length (EL): the average length of the simultaneous CIs.

The coverage probabilities and expected lengths are derived as

$$CP = \sum_{M=1}^{5000} \frac{c^{(M)}(L_{il}^{(M)} \leq \lambda_{il} \leq U_{il}^{(M)})}{5000} \quad and \quad EL = \sum_{M=1}^{5000} \frac{(U_{il}^{(M)} - L_{il}^{(M)})}{5000},$$

where $c^{(M)}(L_{il}^{(M)} \leq \lambda_{il} \leq U_{il}^{(M)})$ is the number of $\lambda_{il}$ that is contained in the interval, $L_{il}^{(M)}$ and $U_{il}^{(M)}$ are the lower and upper bounds of the interval respectively, and $M$ is the total number of simulations that were run for the study.

For each scenario, the best-performing CI has a coverage probability above or close to the nominal confidence level (0.95) and the shortest expected length. The performances of the proposed methods were compared via a Monte Carlo simulation study carried out with the aid of the R statistical software suite. For each set of parameters, 5000 iterations of the simulations were run. In addition, for each parameter combination, 2000 replications of the fiducial and Bayesian methods were performed. Figure 1 show a flowchart for the simulation study. The chosen sample sizes were 30, 50, or 100. As reported in Tables 1 and 2, we used 12 parameter settings for $\delta_{i(1)}$, $\alpha_i$, and $\beta_i = 1$ with $k = 3$ or $k = 6$.

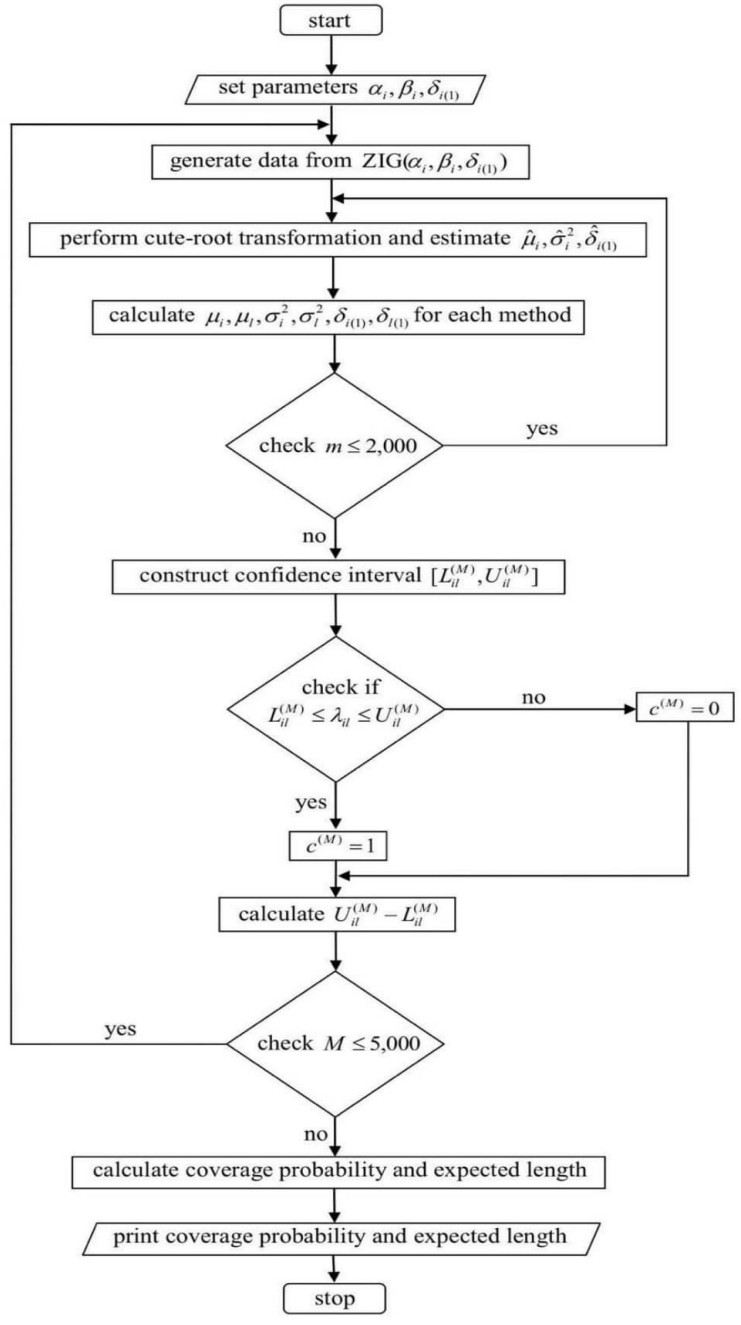

**Figure 1.** A flowchart of the simulation study.

**Table 1.** Simulation study parameter settings for $\delta_{i(1)}$, $\alpha_i$, and $\beta_i = 1$ with $k = 3$.

| Settings | $\delta_{1(1)}$ | $\delta_{2(1)}$ | $\delta_{3(1)}$ | $\alpha_1$ | $\alpha_2$ | $\alpha_3$ |
|---|---|---|---|---|---|---|
| 1 | 0.3 | 0.3 | 0.3 | 1.5 | 1.5 | 1.5 |
| 2 | 0.3 | 0.3 | 0.3 | 2.0 | 2.0 | 2.0 |
| 3 | 0.3 | 0.3 | 0.3 | 2.5 | 2.5 | 2.5 |
| 4 | 0.3 | 0.3 | 0.3 | 3.0 | 3.0 | 3.0 |
| 5 | 0.5 | 0.5 | 0.5 | 2.5 | 2.5 | 2.5 |
| 6 | 0.5 | 0.5 | 0.5 | 3.0 | 3.0 | 3.0 |
| 7 | 0.5 | 0.5 | 0.5 | 3.5 | 3.5 | 3.5 |
| 8 | 0.5 | 0.5 | 0.5 | 4.0 | 4.0 | 4.0 |
| 9 | 0.8 | 0.8 | 0.8 | 5.0 | 5.0 | 5.0 |
| 10 | 0.8 | 0.8 | 0.8 | 5.5 | 5.5 | 5.5 |
| 11 | 0.8 | 0.8 | 0.8 | 6.0 | 6.0 | 6.0 |
| 12 | 0.8 | 0.8 | 0.8 | 6.5 | 6.5 | 6.5 |

**Table 2.** Simulation study parameter settings for $\delta_{i(1)}$, $\alpha_i$, and $\beta_i = 1$ with $k = 6$.

| Settings | $\delta_{1(1)}$ | $\delta_{2(1)}$ | $\delta_{3(1)}$ | $\delta_{4(1)}$ | $\delta_{5(1)}$ | $\delta_{6(1)}$ | $\alpha_1$ | $\alpha_2$ | $\alpha_3$ | $\alpha_4$ | $\alpha_5$ | $\alpha_6$ |
|---|---|---|---|---|---|---|---|---|---|---|---|---|
| 1 | 0.3 | 0.3 | 0.3 | 0.3 | 0.3 | 0.3 | 1.5 | 1.5 | 1.5 | 1.5 | 1.5 | 1.5 |
| 2 | 0.3 | 0.3 | 0.3 | 0.3 | 0.3 | 0.3 | 2.0 | 2.0 | 2.0 | 2.0 | 2.0 | 2.0 |
| 3 | 0.3 | 0.3 | 0.3 | 0.3 | 0.3 | 0.3 | 2.5 | 2.5 | 2.5 | 2.5 | 2.5 | 2.5 |
| 4 | 0.3 | 0.3 | 0.3 | 0.3 | 0.3 | 0.3 | 3.0 | 3.0 | 3.0 | 3.0 | 3.0 | 3.0 |
| 5 | 0.5 | 0.5 | 0.5 | 0.5 | 0.5 | 0.5 | 2.5 | 2.5 | 2.5 | 2.5 | 2.5 | 2.5 |
| 6 | 0.5 | 0.5 | 0.5 | 0.5 | 0.5 | 0.5 | 3.0 | 3.0 | 3.0 | 3.0 | 3.0 | 3.0 |
| 7 | 0.5 | 0.5 | 0.5 | 0.5 | 0.5 | 0.5 | 3.5 | 3.5 | 3.5 | 3.5 | 3.5 | 3.5 |
| 8 | 0.5 | 0.5 | 0.5 | 0.5 | 0.5 | 0.5 | 4.0 | 4.0 | 4.0 | 4.0 | 4.0 | 4.0 |
| 9 | 0.8 | 0.8 | 0.8 | 0.8 | 0.8 | 0.8 | 5.0 | 5.0 | 5.0 | 5.0 | 5.0 | 5.0 |
| 10 | 0.8 | 0.8 | 0.8 | 0.8 | 0.8 | 0.8 | 5.5 | 5.5 | 5.5 | 5.5 | 5.5 | 5.5 |
| 11 | 0.8 | 0.8 | 0.8 | 0.8 | 0.8 | 0.8 | 6.0 | 6.0 | 6.0 | 6.0 | 6.0 | 6.0 |
| 12 | 0.8 | 0.8 | 0.8 | 0.8 | 0.8 | 0.8 | 6.5 | 6.5 | 6.5 | 6.5 | 6.5 | 6.5 |

## 4. Results

### 4.1. Simulation Study

A computer with the AMD Ryzen 3 3250U with Radeon Graphics 8.00 GB of RAM is used to conduct all of the simultaneous CIs. For each program run for all six proposed approaches, we also compare the time consumption for the CIs with various simulation study cases from the coverage probabilities and expected length of the six simultaneous CI methods for $k = 3$ and 6 in Tables 3 and 4, respectively. The coverage probabilities of the Bayesian and HPD interval based on Jeffreys rule or uniform priors were nearly always equal to or greater than the nominal confidence level of 0.95. With settings 4 and 8, the fiducial GCI provided coverage probability greater than 0.95 even though their expected lengths were shorter than the others, while the MOVER were less than the nominal confidence level 0.95 in all case for $k = 3$ and 6. Thus, the simultaneous CIs for the ratios of the means of multiple ZIG distributions cannot be constructed using the methods based on the MOVER. Therefore, the Bayesian and HPD interval based on the Jeffreys rule or uniform priors and the fiducial GCI should be used to compute the simultaneous CIs for the ratios of the means of multiple ZIG distributions, because the CIs which provided the coverage probabilities equal to or greater 0.95. After that, the expected lengths of these CIs are considered to find the shortest length to be the best CI. In almost all settings, we discovered that the expected lengths of HPD intervals based on the Jeffreys rule prior was the smallest length of coverage probabilities over 0.95, while settings 4 and 8 the fiducial GCI was the shortest length. The coverage probabilities and expected lengths of the 95% simultaneous CI methods with various sample sizes are shown in Figures 2 and 3, respectively, while those with various probabilities of nonzero values are displayed in Figures 4 and 5, respectively.

**Table 3.** Coverage probabilities and expected lengths for the 95% simultaneous CIs with $\lambda_{il}(k=3)$.

| Settings | $(n_1, n_2, n_3)$ | Coverage Probability (Expected Length) | | | | | | Time (s) |
|---|---|---|---|---|---|---|---|---|
| | | Fiducial GCI | Baye.Jef | Baye.Uni | HPD.Jef | HPD.Uni | MOVER | |
| 1 | (30,30,30) | 0.9165 | 0.9602 | 0.9743 | **0.9560** | 0.9700 | 0.9426 | 283.39 |
| | | (1.6302) | (1.9560) | (2.2115) | **(1.7946)** | (1.9933) | (2.1646) | |
| | (50,50,50) | 0.9033 | 0.9632 | 0.9703 | **0.9602** | 0.9682 | 0.9432 | 284.09 |
| | | (1.0923) | (1.4044) | (1.4763) | **(1.3293)** | (1.3913) | (1.5019) | |
| | (100,100,100) | 0.8899 | 0.9633 | 0.9666 | **0.9613** | 0.9647 | 0.9505 | 270.73 |
| | | (0.7092) | (0.9389) | (0.9571) | **(0.9101)** | (0.9269) | (1.0092) | |
| | (30,50,100) | 0.9100 | 0.9609 | 0.9717 | **0.9610** | 0.9690 | 0.9342 | 209.83 |
| | | (1.2505) | (1.4883) | (1.6258) | **(1.4097)** | (1.5086) | (1.6496) | |
| 2 | (30,30,30) | 0.9374 | 0.9765 | 0.9854 | **0.9726** | 0.9804 | 0.9266 | 247.25 |
| | | (1.4704) | (1.8333) | (2.0313) | **(1.6922)** | (1.8484) | (1.7522) | |
| | (50,50,50) | 0.9273 | 0.9762 | 0.9812 | **0.9767** | 0.9807 | 0.9310 | 269.32 |
| | | (1.0177) | (1.3464) | (1.4039) | **(1.2778)** | (1.3279) | (1.2522) | |
| | (100,100,100) | 0.9206 | 0.9807 | 0.9827 | **0.9798** | 0.9816 | 0.9386 | 312.92 |
| | | (0.6713) | (0.9094) | (0.9248) | **(0.8826)** | (0.8969) | (0.8529) | |
| | (30,50,100) | 0.9350 | 0.9791 | 0.9864 | **0.9798** | 0.9826 | 0.9207 | 211.88 |
| | | (1.1299) | (1.4033) | (1.4983) | **(1.3389)** | (1.4095) | (1.3463) | |
| 3 | (30,30,30) | **0.9547** | 0.9869 | 0.9921 | 0.9842 | 0.9888 | 0.9156 | 255.33 |
| | | **(1.3820)** | (1.7695) | (1.9318) | (1.6378) | (1.7686) | (1.4955) | |
| | (50,50,50) | 0.9495 | 0.9884 | 0.9908 | **0.9870** | 0.9890 | 0.9256 | 263.27 |
| | | (0.9710) | (1.3080) | (1.3581) | **(1.2435)** | (1.2874) | (1.0849) | |
| | (100,100,100) | 0.9447 | 0.9892 | 0.9903 | **0.9875** | 0.9886 | 0.9345 | 292.09 |
| | | (0.6496) | (0.8912) | (0.9052) | **(0.8655)** | (0.8786) | (0.7496) | |
| | (30,50,100) | **0.9560** | 0.9893 | 0.9916 | 0.9864 | 0.9881 | 0.9196 | 203.86 |
| | | **(1.0559)** | (1.3507) | (1.4220) | (1.2944) | (1.3485) | (1.1532) | |
| 4 | (30,30,30) | **0.9666** | 0.9938 | 0.9960 | 0.9903 | 0.9928 | 0.9083 | 233.40 |
| | | **(1.3205)** | (1.7187) | (1.8600) | (1.5934) | (1.7081) | (1.3274) | |
| | (50,50,50) | **0.9640** | 0.9948 | 0.9954 | 0.9922 | 0.9937 | 0.9150 | 257.29 |
| | | **(0.9448)** | (1.2869) | (1.3323) | (1.2242) | (1.2640) | (0.9768) | |
| | (100,100,100) | **0.9609** | 0.9954 | 0.9953 | 0.9944 | 0.9949 | 0.9240 | 287.83 |
| | | **(0.6374)** | (0.8824) | (0.8948) | (0.8570) | (0.8689) | (0.6773) | |
| | (30,50,100) | **0.9679** | 0.9935 | 0.9954 | 0.9923 | 0.9934 | 0.9072 | 198.29 |
| | | **(1.0204)** | (1.3296) | (1.3894) | (1.2769) | (1.3226) | (1.0352) | |
| 5 | (30,30,30) | 0.9101 | 0.9705 | 0.9764 | **0.9694** | 0.9742 | 0.9302 | 306.95 |
| | | (0.8426) | (1.1082) | (1.1637) | **(1.0645)** | (1.1144) | (1.0624) | |
| | (50,50,50) | 0.9057 | 0.9738 | 0.9775 | **0.9712** | 0.9735 | 0.9402 | 298.94 |
| | | (0.6205) | (0.8350) | (0.8570) | **(0.8130)** | (0.8334) | (0.8101) | |
| | (100,100,100) | 0.8993 | 0.9730 | 0.9746 | **0.9710** | 0.9716 | 0.9435 | 357.10 |
| | | (0.4259) | (0.5801) | (0.5875) | **(0.5706)** | (0.5776) | (0.5748) | |
| | (30,50,100) | 0.9125 | 0.9724 | 0.9764 | **0.9721** | 0.9735 | 0.9308 | 241.91 |
| | | (0.6510) | (0.8515) | (0.8732) | **(0.8349)** | (0.8542) | (0.8332) | |
| 6 | (30,30,30) | 0.9351 | 0.9857 | 0.9892 | **0.9829** | 0.9872 | 0.9233 | 204.06 |
| | | (0.8125) | (1.0845) | (1.1347) | **(1.0428)** | (1.0881) | (0.9480) | |
| | (50,50,50) | 0.9296 | 0.9851 | 0.9875 | **0.9823** | 0.9858 | 0.9318 | 263.42 |
| | | (0.6036) | (0.8221) | (0.8422) | **(0.8009)** | (0.8196) | (0.7264) | |
| | (100,100,100) | 0.9250 | 0.9850 | 0.9864 | **0.9837** | 0.9855 | 0.9379 | 252.54 |
| | | (0.4173) | (0.5741) | (0.5807) | **(0.5647)** | (0.5711) | (0.5198) | |
| | (30,50,100) | 0.9281 | 0.9816 | 0.9833 | **0.9800** | 0.9800 | 0.9200 | 204.76 |
| | | (0.6299) | (0.8361) | (0.8548) | **(0.8208)** | (0.8377) | (0.7459) | |

**Table 3.** *Cont.*

| Settings | $(n_1, n_2, n_3)$ | Coverage Probability (Expected Length) | | | | | | Time (s) |
|---|---|---|---|---|---|---|---|---|
| | | Fiducial GCI | Baye.Jef | Baye.Uni | HPD.Jef | HPD.Uni | MOVER | |
| 7 | (30,30,30) | 0.9467 | 0.9894 | 0.9919 | **0.9874** | 0.9898 | 0.9156 | 308.69 |
| | | (0.7920) | (1.0685) | (1.1148) | **(1.0279)** | (1.0700) | (0.8608) | |
| | (50,50,50) | 0.9454 | 0.9902 | 0.9914 | **0.9881** | 0.9898 | 0.9188 | 220.02 |
| | | (0.5940) | (0.8153) | (0.8348) | **(0.7944)** | (0.8126) | (0.6665) | |
| | (100,100,100) | 0.9388 | 0.9886 | 0.9898 | **0.9884** | 0.9884 | 0.9316 | 317.59 |
| | | (0.4111) | (0.5694) | (0.5754) | **(0.5602)** | (0.5659) | (0.4782) | |
| | (30,50,100) | 0.9472 | 0.9890 | 0.9906 | **0.9872** | 0.9865 | 0.9173 | 194.66 |
| | | (0.6137) | (0.8236) | (0.8397) | **(0.8089)** | (0.8237) | (0.6842) | |
| 8 | (30,30,30) | **0.9609** | 0.9934 | 0.9955 | 0.9920 | 0.9942 | 0.9089 | 277.01 |
| | | **(0.7804)** | (1.0601) | (1.1047) | (1.0204) | (1.0607) | (0.8020) | |
| | (50,50,50) | **0.9559** | 0.9945 | 0.9956 | 0.9936 | 0.9944 | 0.9172 | 270.93 |
| | | **(0.5865)** | (0.8093) | (0.8280) | (0.7887) | (0.8061) | (0.6173) | |
| | (100,100,100) | **0.9559** | 0.9947 | 0.9955 | 0.9936 | 0.9948 | 0.9303 | 347.70 |
| | | **(0.4065)** | (0.5652) | (0.5717) | (0.5561) | (0.5623) | (0.4445) | |
| | (30,50,100) | **0.9588** | 0.9924 | 0.9932 | 0.9905 | 0.9910 | 0.9082 | 207.91 |
| | | **(0.6049)** | (0.8187) | (0.8332) | (0.8046) | (0.8181) | (0.6339) | |
| 9 | (30,30,30) | 0.8701 | 0.9589 | 0.9666 | **0.9557** | 0.9655 | 0.9268 | 230.70 |
| | | (0.3935) | (0.5351) | (0.5624) | **(0.5266)** | (0.5531) | (0.5321) | |
| | (50,50,50) | 0.8590 | 0.9562 | 0.9637 | **0.9534** | 0.9604 | 0.9314 | 306.28 |
| | | (0.2979) | (0.4088) | (0.4213) | **(0.4040)** | (0.4161) | (0.4205) | |
| | (100,100,100) | 0.8590 | 0.9556 | 0.9591 | **0.9535** | 0.9576 | 0.9456 | 257.02 |
| | | (0.2070) | (0.2858) | (0.2901) | **(0.2832)** | (0.2874) | (0.3046) | |
| | (30,50,100) | 0.8585 | 0.9493 | 0.9552 | **0.9573** | 0.9524 | 0.9261 | 263.90 |
| | | (0.3057) | (0.4104) | (0.4228) | **(0.4074)** | (0.4196) | (0.4235) | |
| 10 | (30,30,30) | 0.8792 | 0.9627 | 0.9712 | **0.9624** | 0.9707 | 0.9164 | 205.61 |
| | | (0.3906) | (0.5338) | (0.5607) | **(0.5253)** | (0.5514) | (0.5066) | |
| | (50,50,50) | 0.8777 | 0.9652 | 0.9716 | **0.9632** | 0.9690 | 0.9306 | 239.42 |
| | | (0.2951) | (0.4068) | (0.4192) | **(0.4020)** | (0.4141) | (0.3979) | |
| | (100,100,100) | 0.8672 | 0.9630 | 0.9656 | **0.9602** | 0.9653 | 0.9409 | 348.64 |
| | | (0.2056) | (0.2847) | (0.2892) | **(0.2821)** | (0.2865) | (0.2894) | |
| | (30,50,100) | 0.8762 | 0.9602 | 0.9635 | **0.9575** | 0.9612 | 0.9234 | 308.76 |
| | | (0.3024) | (0.4086) | (0.4205) | **(0.4056)** | (0.4173) | (0.4017) | |
| 11 | (30,30,30) | 0.8946 | 0.9718 | 0.9778 | **0.9703** | 0.9768 | 0.9126 | 209.98 |
| | | (0.3865) | (0.5305) | (0.5570) | **(0.5221)** | (0.5477) | (0.4789) | |
| | (50,50,50) | 0.8875 | 0.9710 | 0.9750 | **0.9700** | 0.9740 | 0.9276 | 299.44 |
| | | (0.2938) | (0.4064) | (0.4186) | **(0.4016)** | (0.4136) | (0.3796) | |
| | (100,100,100) | 0.8815 | 0.9698 | 0.9728 | **0.9681** | 0.9706 | 0.9354 | 283.05 |
| | | (0.2041) | (0.2839) | (0.2881) | **(0.2813)** | (0.2855) | (0.2756) | |
| | (30,50,100) | 0.8916 | 0.9683 | 0.9706 | **0.9666** | 0.9687 | 0.9222 | 285.78 |
| | | (0.3000) | (0.4069) | (0.4186) | **(0.4039)** | (0.4155) | (0.3833) | |
| 12 | (30,30,30) | 0.9056 | 0.9750 | 0.9816 | **0.9747** | 0.9802 | 0.9129 | 214.65 |
| | | (0.3847) | (0.5293) | (0.5560) | **(0.5209)** | (0.5467) | (0.4609) | |
| | (50,50,50) | 0.9020 | 0.9777 | 0.9818 | **0.9763** | 0.9809 | 0.9264 | 281.94 |
| | | (0.2917) | (0.4045) | (0.4166) | **(0.3997)** | (0.4116) | (0.3636) | |
| | (100,100,100) | 0.8956 | 0.9736 | 0.9751 | **0.9722** | 0.9728 | 0.9303 | 286.10 |
| | | (0.2030) | (0.2829) | (0.2872) | **(0.2804)** | (0.2846) | (0.2639) | |
| | (30,50,100) | 0.8984 | 0.9750 | 0.9758 | **0.9727** | 0.9748 | 0.9186 | 302.18 |
| | | (0.2977) | (0.4051) | (0.4166) | **(0.4021)** | (0.4136) | (0.3660) | |

Note: Bold denotes the best-performing method.

**Table 4.** Coverage probabilities and expected lengths for the 95% simultaneous CIs with $\lambda_{il}(k=6)$.

| Settings | $(n_1, n_2, n_3, n_4, n_5, n_6)$ | Coverage Probability (Expected Length) | | | | | | Time (s) |
|---|---|---|---|---|---|---|---|---|
| | | Fiducial GCI | Baye.Jef | Baye.Uni | HPD.Jef | HPD.Uni | MOVER | |
| 1 | (30,30,30,30,30,30) | 0.9183 | 0.9606 | 0.9754 | **0.9581** | 0.9712 | 0.9380 | 457.73 |
| | | (1.6233) | (1.9504) | (2.2040) | **(1.7896)** | (1.9867) | (2.1529) | |
| | (50,50,50,50,50,50) | 0.8991 | 0.9604 | 0.9681 | **0.9577** | 0.9644 | 0.9419 | 497.40 |
| | | (1.0906) | (1.4020) | (1.4737) | **(1.3268)** | (1.3888) | (1.5002) | |
| | (100,100,100,100,100,100) | 0.8904 | 0.9633 | 0.9664 | **0.9609** | 0.9642 | 0.9511 | 432.44 |
| | | (0.7105) | (0.9404) | (0.9588) | **(0.9115)** | (0.9285) | (1.0127) | |
| | (30,30,50,50,100,100) | 0.9100 | 0.9616 | 0.9713 | **0.9630** | 0.9702 | 0.9391 | 543.09 |
| | | (1.2243) | (1.4732) | (1.5988) | **(1.3935)** | (1.4851) | (1.6181) | |
| 2 | (30,30,30,30,30,30) | 0.9392 | 0.9770 | 0.9861 | **0.9748** | 0.9824 | 0.9288 | 493.07 |
| | | (1.4757) | (1.8416) | (2.0410) | **(1.6993)** | (1.8576) | (1.7567) | |
| | (50,50,50,50,50,50) | 0.9270 | 0.9787 | 0.9830 | **0.9758** | 0.9802 | 0.9292 | 485.92 |
| | | (1.0121) | (1.3392) | (1.3966) | **(1.2713)** | (1.3214) | (1.2424) | |
| | (100,100,100,100,100,100) | 0.9231 | 0.9813 | 0.9824 | **0.9792** | 0.9812 | 0.9421 | 471.38 |
| | | (0.6708) | (0.9089) | (0.9247) | **(0.8821)** | (0.8967) | (0.8537) | |
| | (30,30,50,50,100,100) | 0.9362 | 0.9800 | 0.9852 | **0.9795** | 0.9830 | 0.9311 | 545.98 |
| | | (1.1152) | (1.3976) | (1.4871) | **(1.3305)** | (1.3977) | (1.3345) | |
| 3 | (30,30,30,30,30,30) | **0.9557** | 0.9877 | 0.9922 | 0.9843 | 0.9890 | 0.9177 | 433.06 |
| | | **(1.3864)** | (1.7734) | (1.9379) | (1.6413) | (1.7738) | (1.5065) | |
| | (50,50,50,50,50,50) | 0.9483 | 0.9895 | 0.9915 | **0.9866** | 0.9893 | 0.9242 | 534.74 |
| | | (0.9687) | (1.3043) | (1.3540) | **(1.2399)** | (1.2834) | (1.0823) | |
| | (100,100,100,100,100,100) | 0.9436 | 0.9897 | 0.9909 | **0.9873** | 0.9886 | 0.9323 | 549.42 |
| | | (0.6498) | (0.8920) | (0.9063) | **(0.8662)** | (0.8795) | (0.7505) | |
| | (30,30,50,50,100,100) | **0.9503** | 0.9880 | 0.9907 | 0.9862 | 0.9882 | 0.9171 | 419.67 |
| | | **(1.0499)** | (1.3507) | (1.4200) | (1.2905) | (1.3436) | (1.1514) | |
| 4 | (30,30,30,30,30,30) | **0.9694** | 0.9938 | 0.9961 | 0.9911 | 0.9940 | 0.9104 | 480.74 |
| | | **(1.3289)** | (1.7301) | (1.8733) | (1.6042 | (1.7203) | (1.3388) | |
| | (50,50,50,50,50,50) | **0.9634** | 0.9939 | 0.9953 | 0.9926 | 0.9940 | 0.9162 | 427.10 |
| | | **(0.9445)** | (1.2864) | (1.3321) | (1.2239) | (1.2640) | (0.9716) | |
| | (100,100,100,100,100,100) | **0.9622** | 0.9954 | 0.9959 | 0.9941 | 0.9948 | 0.9257 | 522.90 |
| | | **(0.6384)** | (0.8833) | (0.8966) | (0.8580) | (0.8704) | (0.6790) | |
| | (30,30,50,50,100,100) | **0.9664** | 0.9938 | 0.9953 | 0.9924 | 0.9932 | 0.9090 | 494.16 |
| | | **(1.0107)** | (1.3246) | (1.3814) | (1.2683) | (1.3124) | (1.0272) | |
| 5 | (30,30,30,30,30,30) | 0.9084 | 0.9709 | 0.9770 | **0.9672** | 0.9747 | 0.9253 | 561.27 |
| | | (0.8430) | (1.1089) | (1.1648) | **(1.0653)** | (1.1155) | (1.0647) | |
| | (50,50,50,50,50,50) | 0.9056 | 0.9742 | 0.9778 | **0.9721** | 0.9756 | 0.9374 | 540.49 |
| | | (0.6213) | (0.8367) | (0.8585) | **(0.8146)** | (0.8350) | (0.8103) | |
| | (100,100,100,100,100,100) | 0.9016 | 0.9744 | 0.97587 | **0.9726** | 0.9738 | 0.9472 | 426.37 |
| | | (0.4256) | (0.5800) | (0.5870) | **(0.5704)** | (0.5771) | (0.5747) | |
| | (30,30,50,50,100,100) | 0.9100 | 0.9729 | 0.9762 | **0.9706** | 0.9732 | 0.9309 | 572.72 |
| | | (0.6443) | (0.8466) | (0.8680) | **(0.8283)** | (0.8475) | (0.8255) | |
| 6 | (30,30,30,30,30,30) | 0.9326 | 0.9826 | 0.9866 | **0.9801** | 0.9842 | 0.9223 | 485.01 |
| | | (0.8131) | (1.0861) | (1.1363) | **(1.0442)** | (1.0897) | (0.9503) | |
| | (50,50,50,50,50,50) | 0.9270 | 0.9841 | 0.9865 | **0.9823** | 0.9843 | 0.9322 | 539.26 |
| | | (0.6044) | (0.8232) | (0.8434) | **(0.8017)** | (0.8208) | (0.7279) | |
| | (100,100,100,100,100,100) | 0.9236 | 0.9854 | 0.9863 | **0.9838** | 0.9848 | 0.9387 | 430.47 |
| | | (0.4170) | (0.5740) | (0.5805) | **(0.5647)** | (0.5708) | (0.5198) | |
| | (30,30,50,50,100,100) | 0.9291 | 0.9831 | 0.9850 | **0.9812** | 0.9822 | 0.9244 | 584.41 |
| | | (0.6263) | (0.8348) | (0.8537) | **(0.8177)** | (0.8349) | (0.7451) | |

**Table 4.** *Cont.*

| Settings | $(n_1, n_2, n_3, n_4, n_5, n_6)$ | Coverage Probability (Expected Length) | | | | | | Time (s) |
|---|---|---|---|---|---|---|---|---|
| | | Fiducial GCI | Baye.Jef | Baye.Uni | HPD.Jef | HPD.Uni | MOVER | |
| 7 | (30,30,30,30,30,30) | 0.9466 | 0.9890 | 0.9918 | **0.9871** | 0.9899 | 0.9145 | 441.96 |
| | | (0.7945) | (1.0719) | (1.1189) | **(1.0312)** | (1.0738) | (0.8652) | |
| | (50,50,50,50,50,50) | 0.9451 | 0.9902 | 0.9917 | **0.9890** | 0.9904 | 0.9261 | 474.42 |
| | | (0.5938) | (0.8150) | (0.8344) | **(0.7941)** | (0.8121) | (0.6662) | |
| | (100,100,100,100,100,100) | 0.9412 | 0.9900 | 0.9904 | **0.9884** | 0.9892 | 0.9346 | 493.57 |
| | | (0.4106) | (0.5684) | (0.5747) | **(0.5592)** | (0.5652) | (0.4772) | |
| | (30,30,50,50,100,100) | 0.9454 | 0.9900 | 0.9914 | **0.9889** | 0.9895 | 0.9200 | 452.55 |
| | | (0.6122) | (0.8242) | (0.8413) | **(0.8078)** | (0.8234) | (0.6809) | |
| 8 | (30,30,30,30,30,30) | **0.9569** | 0.9931 | 0.9950 | 0.9912 | 0.9929 | 0.9126 | 445.08 |
| | | **(0.7807)** | (1.0609) | (1.1057) | (1.0211) | (1.0617) | (0.7992) | |
| | (50,50,50,50,50,50) | **0.9569** | 0.9941 | 0.9951 | 0.9929 | 0.9938 | 0.9191 | 538.73 |
| | | **(0.5865)** | (0.8093) | (0.8282) | (0.7887) | (0.8063) | (0.6195) | |
| | (100,100,100,100,100,100) | **0.9535** | 0.9937 | 0.9943 | 0.9926 | 0.9933 | 0.9271 | 430.94 |
| | | **(0.4073)** | (0.5661) | (0.5725) | (0.5570) | (0.5632) | (0.4445) | |
| | (30,30,50,50,100,100) | **0.9563** | 0.9921 | 0.9933 | 0.9907 | 0.9910 | 0.9100 | 620.11 |
| | | **(0.6021)** | (0.8171) | (0.8324) | (0.8012) | (0.8152) | (0.6300) | |
| 9 | (30,30,30,30,30,30) | 0.8660 | 0.9564 | 0.9662 | **0.9549** | 0.9645 | 0.9241 | 496.95 |
| | | (0.3942) | (0.5361) | (0.5632) | **(0.5275)** | (0.5538) | (0.5322) | |
| | (50,50,50,50,50,50) | 0.8614 | 0.9558 | 0.9623 | **0.9547** | 0.9602 | 0.9346 | 461.39 |
| | | (0.2976) | (0.4088) | (0.4213) | **(0.4039)** | (0.4161) | (0.4195) | |
| | (100,100,100,100,100,100) | 0.8516 | 0.9523 | 0.9558 | **0.9502** | 0.9533 | 0.9414 | 445.27 |
| | | (0.2069) | (0.2856) | (0.2901) | **(0.2830)** | (0.2874) | (0.3047) | |
| | (30,30,50,50,100,100) | 0.8630 | 0.9540 | 0.9584 | **0.9515** | 0.9564 | 0.9311 | 514.16 |
| | | (0.3045) | (0.4108) | (0.4229) | **(0.4073)** | (0.4193) | (0.4222) | |
| 10 | (30,30,30,30,30,30) | 0.8786 | 0.9638 | 0.9724 | **0.9615** | 0.9702 | 0.9171 | 552.72 |
| | | (0.3903) | (0.5333) | (0.5600) | **(0.5249)** | (0.5506) | (0.5045) | |
| | (50,50,50,50,50,50) | 0.8754 | 0.9642 | 0.9695 | **0.9623** | 0.9676 | 0.9305 | 560.22 |
| | | (0.2953) | (0.4071) | (0.4195) | **(0.4022)** | (0.4144) | (0.3983) | |
| | (100,100,100,100,100,100) | 0.8695 | 0.9628 | 0.9651 | **0.9610** | 0.9638 | 0.9393 | 414.03 |
| | | (0.2052) | (0.2843) | (0.2887) | **(0.2817)** | (0.2860) | (0.2889) | |
| | (30,30,50,50,100,100) | 0.8758 | 0.9623 | 0.9652 | **0.9609** | 0.9634 | 0.9253 | 480.58 |
| | | (0.3011) | (0.4083) | (0.4203) | **(0.4048)** | (0.4167) | (0.4000) | |
| 11 | (30,30,30,30,30,30) | 0.8926 | 0.9716 | 0.9784 | **0.9690** | 0.9767 | 0.9183 | 504.72 |
| | | (0.3868) | (0.5306) | (0.5569) | **(0.5222)** | (0.5476) | (0.4806) | |
| | (50,50,50,50,50,50) | 0.8893 | 0.9724 | 0.9774 | **0.9706** | 0.9747 | 0.9276 | 514.94 |
| | | (0.2928) | (0.4051) | (0.4173) | **(0.4003)** | (0.4123) | (0.3797) | |
| | (100,100,100,100,100,100) | 0.8824 | 0.9682 | 0.9706 | **0.9666** | 0.9692 | 0.9378 | 427.04 |
| | | (0.2042) | (0.2837) | (0.2881) | **(0.2811)** | (0.2855) | (0.2761) | |
| | (30,30,50,50,100,100) | 0.8913 | 0.9697 | 0.9727 | **0.9679** | 0.9705 | 0.9232 | 460.11 |
| | | (0.2989) | (0.4068) | (0.4189) | **(0.4034)** | (0.4152) | (0.3817) | |
| 12 | (30,30,30,30,30,30) | 0.9023 | 0.9770 | 0.9824 | **0.9750** | 0.9810 | 0.9137 | 482.03 |
| | | (0.3843) | (0.5289) | (0.5553) | **(0.5206)** | (0.5460) | (0.4600) | |
| | (50,50,50,50,50,50) | 0.8995 | 0.9769 | 0.9804 | **0.9751** | 0.9790 | 0.9240 | 453.36 |
| | | (0.2914) | (0.4042) | (0.4164) | **(0.3994)** | (0.4114) | (0.3633) | |
| | (100,100,100,100,100,100) | 0.8965 | 0.9769 | 0.9797 | **0.9759** | 0.9777 | 0.9323 | 423.01 |
| | | (0.2031) | (0.2829) | (0.2873) | **(0.2803)** | (0.2847) | (0.2646) | |
| | (30,30,50,50,100,100) | 0.9015 | 0.9750 | 0.9770 | **0.9737** | 0.9754 | 0.9178 | 521.84 |
| | | (0.2970) | (0.4056) | (0.4175) | **(0.4022)** | (0.4139) | (0.3659) | |

Note: Bold denotes the best-performing method.

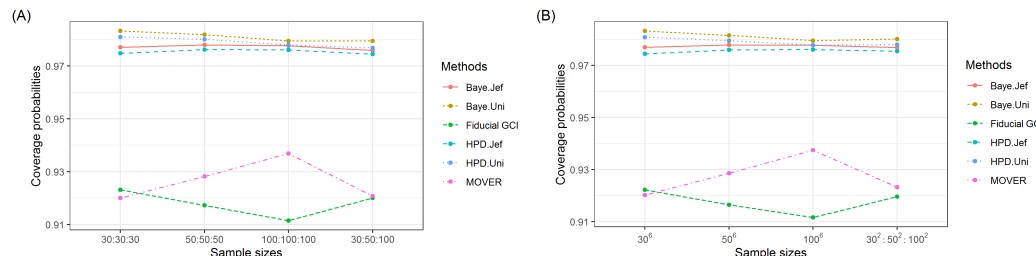

**Figure 2.** Coverage probabilities of the 95% simultaneous CIs with various sample sizes: (**A**) $k = 3$ and (**B**) $k = 6$.

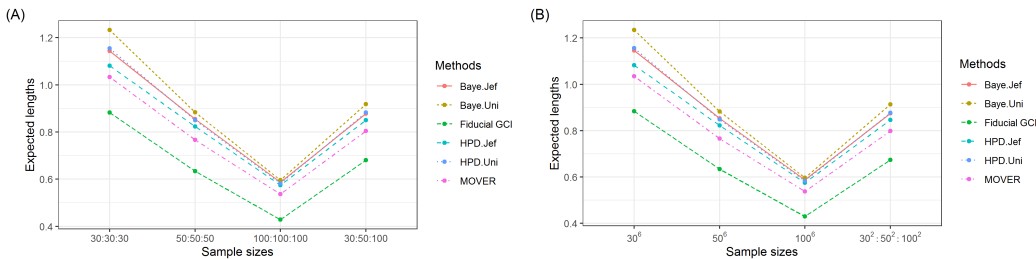

**Figure 3.** Expected lengths of the 95% simultaneous CIs with various sample sizes: (**A**) $k = 3$ and (**B**) $k = 6$.

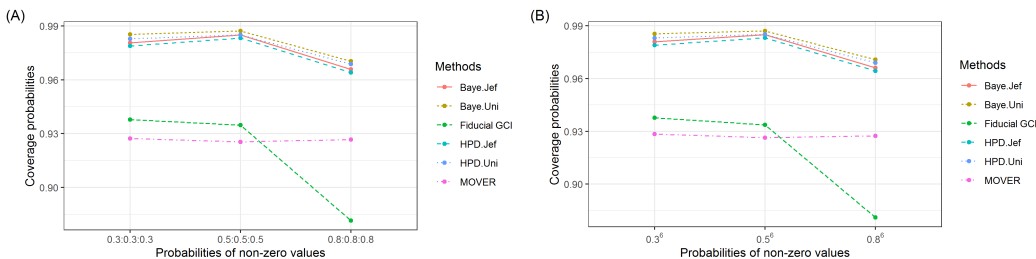

**Figure 4.** Coverage probabilities of the 95% simultaneous CIs with various probabilities of nonzero values: (**A**) $k = 3$ and (**B**) $k = 6$.

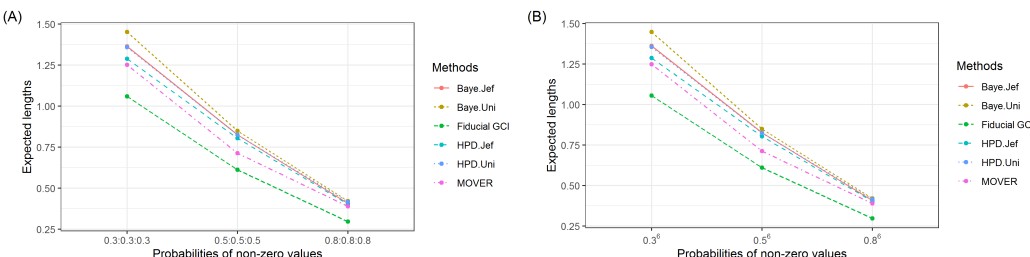

**Figure 5.** Expected lengths of the 95% simultaneous CIs with various probabilities of nonzero values: (**A**) $k = 3$ and (**B**) $k = 6$.

*4.2. Empirical Application of the Simultaneous CI Methods to Rainfall Data in Thailand*

The study of Kaewprasert et al. [1] was utilized to estimate rainfall data from ZIG distributions. Thailand was separated into six regions, from which rainfall datasets for September 2021 from the following rain stations were used in this analysis as shown in Table 5:

1. Northern (R1): Chiang Mai [22].
2. Southern (R2): Trang [23].
3. Northeastern (R3): Chaiyaphum [24].

4. Eastern (R4): Prachin Buri [25].
5. Western (R5): Kanchanaburi [26].
6. Central (R6): Kamphaeng Phet [27].

**Table 5.** The daily rainfall data for September 2021 in Thailand by region.

| Region | Daily Rainfall (mm) | | | | | | | | | |
|---|---|---|---|---|---|---|---|---|---|---|
| R1 | 0.2 | 6.5 | 14.5 | 0.0 | 0.0 | 79.0 | 0.0 | 5.0 | 44.5 | 40.0 |
| | 20.0 | 3.6 | 4.2 | 16.0 | 47.7 | 26.7 | 10.1 | 3.0 | 0.0 | 4.1 |
| | 20.2 | 27.8 | 0.0 | 13.0 | 63.5 | 50.0 | 25.0 | 0.0 | 7.3 | 2.4 |
| R2 | 22.5 | 2.0 | 0.0 | 0.0 | 0.0 | 0.0 | 15.5 | 22.5 | 55.0 | 2.6 |
| | 0.3 | 9.6 | 12.0 | 9.7 | 3.2 | 0.0 | 21.2 | 10.3 | 14.2 | 23.4 |
| | 34.6 | 10.5 | 0.0 | 0.8 | 2.0 | 2.6 | 0.5 | 69.4 | 36.8 | 20.5 |
| R3 | 14.4 | 0.5 | 22.5 | 4.6 | 20.5 | 2.0 | 0.0 | 0.0 | 24.2 | 24.4 |
| | 0.4 | 12.0 | 8.5 | 1.9 | 10.0 | 2.5 | 0.0 | 0.0 | 16.4 | 0.0 |
| | 23.4 | 11.5 | 8.1 | 76.2 | 30.3 | 22.9 | 2.0 | 1.8 | 13.0 | 0.0 |
| R4 | 1.3 | 10.2 | 4.2 | 40.5 | 4.9 | 4.0 | 43.4 | 20.3 | 16.2 | 5.6 |
| | 0.0 | 0.0 | 4.5 | 7.4 | 9.1 | 0.6 | 0.0 | 0.2 | 8.3 | 15.7 |
| | 0.0 | 5.6 | 4.3 | 22.4 | 39.1 | 0.0 | 0.0 | 8.2 | 12.1 | 0.0 |
| R5 | 28.1 | 11.5 | 4.2 | 3.4 | 0.0 | 3.4 | 4.3 | 0.2 | 0.0 | 2.9 |
| | 10.7 | 0.0 | 1.3 | 0.0 | 9.1 | 15.9 | 0.0 | 0.8 | 5.2 | 15.1 |
| | 32.1 | 3.9 | 8.9 | 2.6 | 15.1 | 18.1 | 4.0 | 0.0 | 1.0 | 0.0 |
| R6 | 2.5 | 5.5 | 31.5 | 29.5 | 0.0 | 2.0 | 2.5 | 0.0 | 6.0 | 1.0 |
| | 1.5 | 0.0 | 6.0 | 0.0 | 35.5 | 21.0 | 2.5 | 0.5 | 19.0 | 42.0 |
| | 23.0 | 34.0 | 11.5 | 0.5 | 110.5 | 39.0 | 0.0 | 0.0 | 0.0 | 0.0 |

Figure 6 presents the distribution of these data and displays the right-skewness of the daily rainfall datasets for the six regions. We used the minimum Akaike information criterion (AIC) to test the fit of various distributions to the positive rainfall datasets, which is defined as follows:

$$\text{AIC} = -2 \ln L + 2h,$$

where h is the number of parameters and L is the likelihood function. The findings in Table 6 demonstrate that the gamma distribution was the best fit for all of the positive rainfall datasets. Moreover, Figure 7 displays Q-Q plots of the positive daily rainfall datasets, which confirm that they all follow a gamma distribution.

The parameter estimations were computed for the rainfall from six regions as shown in Table 7. The 95% simultaneous CIs for the daily rainfall dataset from six regions of Thailand in September 2021 are reported in Table 8. In accordance with the simulation results in the previous section, the length of the HPD interval based on the Jeffreys rule prior was the most suitable, thereby confirming its suitability for constructing the simultaneous CIs for the ratio of the means of multiple ZIG distributions.

**Table 6.** AIC results to check the distributions of the positive daily rainfall data.

| Distribution | AIC Value | | | | | |
|---|---|---|---|---|---|---|
| | R1 | R2 | R3 | R4 | R5 | R6 |
| Normal | 218.2205 | 208.7326 | 204.8500 | 185.1858 | 167.2802 | 206.9620 |
| Lognormal | 205.5462 | 190.3452 | 181.1234 | 169.9121 | 152.2575 | 176.9593 |
| Cauchy | 221.9773 | 208.2171 | 201.1505 | 179.0343 | 167.1753 | 201.9837 |
| Gamma | 200.9070 | 186.5715 | 179.1330 | 166.2781 | 149.8757 | 175.7948 |
| Logistic | 217.8841 | 206.0280 | 198.2298 | 183.1289 | 165.8773 | 201.3254 |
| t | 219.9017 | 206.4851 | 197.5618 | 180.6719 | 167.3370 | 200.8800 |
| Chi-squared | 365.4868 | 319.1705 | 280.2808 | 230.0381 | 182.1760 | 356.7011 |

**Table 7.** Parameter estimates for the six regions in Thailand.

| Region | $n_i$ | $\hat{\delta}_{i(1)}$ | $\hat{\alpha}_i$ | $\hat{\beta}_i$ | $\hat{\mu}_i$ | $\hat{\sigma}_i^2$ | $\hat{\lambda}_i$ |
|--------|-------|------------------------|------------------|-----------------|---------------|---------------------|-------------------|
| R1 | 30 | 0.80 | 6.04 | 2.41 | 2.50 | 0.91 | 18.02 |
| R2 | 30 | 0.80 | 5.03 | 2.25 | 2.22 | 0.87 | 13.56 |
| R3 | 30 | 0.80 | 5.50 | 2.60 | 2.11 | 0.76 | 11.46 |
| R4 | 30 | 0.77 | 6.61 | 3.18 | 2.07 | 0.58 | 9.66 |
| R5 | 30 | 0.77 | 7.02 | 3.80 | 1.84 | 0.45 | 6.77 |
| R6 | 30 | 0.73 | 4.24 | 1.88 | 2.24 | 1.17 | 14.18 |

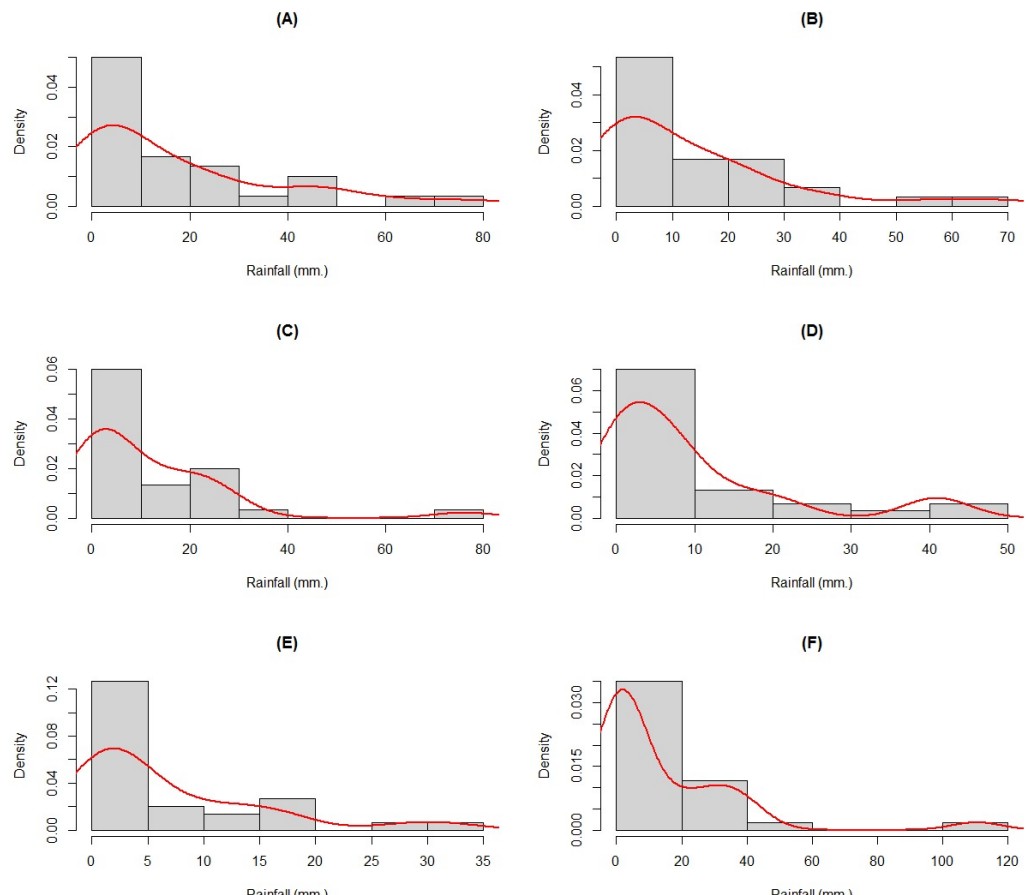

**Figure 6.** The densities of the rainfall datasets for the six regions in Thailand: (**A**) Northern (**B**) Southern (**C**) Northeastern (**D**) Eastern (**E**) Western (**F**) Central.

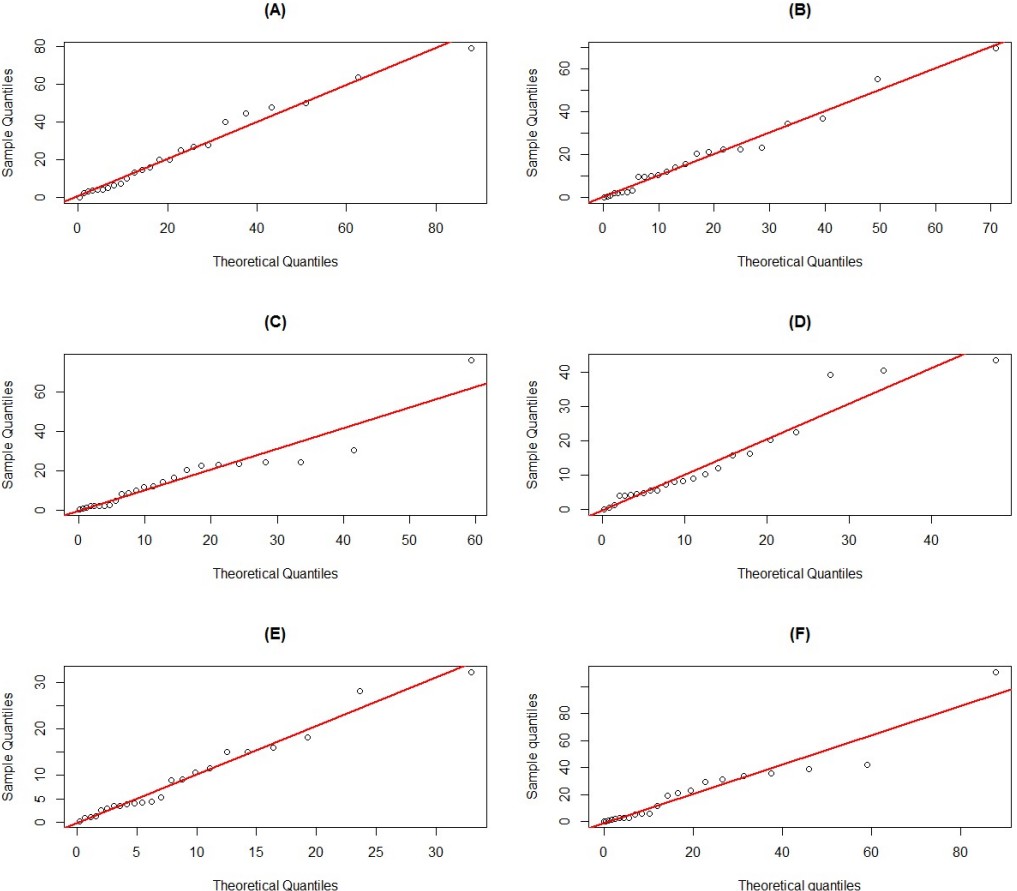

**Figure 7.** Q-Q plots of the nonzero part of the daily rainfall datasets from the six regions in Thailand: (**A**) Northern (**B**) Southern (**C**) Northeastern (**D**) Eastern (**E**) Western (**F**) Central.

**Table 8.** The ratios of the means of the daily rainfall datasets for September 2021 from six regions in Thailand with nominal 95% simultaneous CIs.

| Comparisons | Fiducial GCI | | | Baye.Jef | | | Baye.Uni | | | HPD.Jef | | | HPD.Uni | | | MOVER | | |
|---|---|---|---|---|---|---|---|---|---|---|---|---|---|---|---|---|---|---|
| | Lower | Upper | Length | Lower | Upper | Length | Lower | Upper | Length | Lower | Upper | Length | Lower | Upper | Length | Lower | Upper | Length |
| R1/R2 | 0.9040 | 1.8757 | 0.9718 | 0.8589 | 1.9924 | 1.1335 | 0.8372 | 2.0129 | 1.1757 | 0.8026 | 1.8847 | 1.0820 | 0.8106 | 1.9693 | 1.1587 | 0.6092 | 2.5053 | 1.8961 |
| R1/R3 | 1.1166 | 2.2003 | 1.0837 | 1.0109 | 2.3655 | 1.3546 | 1.0037 | 2.3999 | 1.3963 | 0.9668 | 2.2466 | 1.2798 | 0.9789 | 2.3602 | 1.3813 | 0.7461 | 3.0199 | 2.2738 |
| R1/R4 | 1.3216 | 2.6334 | 1.3118 | 1.2689 | 2.6918 | 1.4230 | 1.2289 | 2.8345 | 1.6056 | 1.2299 | 2.6360 | 1.4060 | 1.1582 | 2.7093 | 1.5511 | 0.9295 | 3.5430 | 2.6135 |
| R1/R5 | 1.9145 | 3.7167 | 1.8023 | 1.7490 | 3.8849 | 2.1358 | 1.7541 | 3.9409 | 2.1868 | 1.6744 | 3.7701 | 2.0957 | 1.6406 | 3.7805 | 2.1399 | 1.3383 | 5.0636 | 3.7253 |
| R1/R6 | 0.8360 | 1.8253 | 0.9893 | 0.8090 | 1.9430 | 1.1340 | 0.7563 | 1.9498 | 1.1935 | 0.7698 | 1.8613 | 1.0916 | 0.7111 | 1.8740 | 1.1629 | 0.5473 | 2.5754 | 2.0281 |
| R2/R3 | 0.8228 | 1.7424 | 0.9197 | 0.7836 | 1.7664 | 0.9827 | 0.7537 | 1.8064 | 1.0526 | 0.7794 | 1.7516 | 0.9722 | 0.7201 | 1.7555 | 1.0354 | 0.5805 | 2.5399 | 1.9594 |
| R2/R4 | 0.9701 | 2.0173 | 1.0472 | 0.9323 | 2.0646 | 1.1323 | 0.9242 | 2.1853 | 1.2611 | 0.8857 | 1.9811 | 1.0954 | 0.8886 | 2.0863 | 1.1977 | 0.7249 | 2.9842 | 2.2593 |
| R2/R5 | 1.4119 | 2.8559 | 1.4440 | 1.3331 | 2.9510 | 1.6179 | 1.2885 | 3.0643 | 1.7758 | 1.2804 | 2.8237 | 1.5434 | 1.2557 | 2.9678 | 1.7120 | 1.0441 | 4.2642 | 3.2201 |
| R2/R6 | 0.6226 | 1.4222 | 0.7995 | 0.5916 | 1.4862 | 0.8947 | 0.5656 | 1.4844 | 0.9189 | 0.5789 | 1.4478 | 0.8690 | 0.5512 | 1.4502 | 0.8990 | 0.4243 | 2.1507 | 1.7264 |
| R3/R4 | 0.8275 | 1.6981 | 0.8706 | 0.8017 | 1.7538 | 0.9521 | 0.7839 | 1.7946 | 1.0107 | 0.7799 | 1.7116 | 0.9317 | 0.7612 | 1.7558 | 0.9946 | 0.6018 | 2.4342 | 1.8324 |
| R3/R5 | 1.1965 | 2.3640 | 1.1675 | 1.1355 | 2.4876 | 1.3521 | 1.1235 | 2.5914 | 1.4678 | 1.0843 | 2.4185 | 1.3342 | 1.0611 | 2.4607 | 1.3996 | 0.8666 | 3.4784 | 2.6118 |
| R3/R6 | 0.5256 | 1.1701 | 0.6444 | 0.5015 | 1.2602 | 0.7587 | 0.4809 | 1.2962 | 0.8153 | 0.4598 | 1.1947 | 0.7349 | 0.4363 | 1.2085 | 0.7723 | 0.3533 | 1.7585 | 1.4051 |
| R4/R5 | 1.0225 | 2.0077 | 0.9852 | 0.9767 | 2.0685 | 1.0919 | 0.9266 | 2.1553 | 1.2288 | 0.9451 | 2.0213 | 1.0762 | 0.9048 | 2.0736 | 1.1688 | 0.7385 | 2.7872 | 2.0488 |
| R4/R6 | 0.4439 | 0.9873 | 0.5434 | 0.4303 | 1.0594 | 0.6291 | 0.3974 | 1.0325 | 0.6351 | 0.3871 | 0.9932 | 0.6061 | 0.3793 | 1.0057 | 0.6265 | 0.3005 | 1.4145 | 1.1140 |
| R5/R6 | 0.3160 | 0.6915 | 0.3755 | 0.3034 | 0.7365 | 0.4331 | 0.2867 | 0.7564 | 0.4697 | 0.2906 | 0.7127 | 0.4221 | 0.2752 | 0.7307 | 0.4555 | 0.2103 | 0.9829 | 0.7726 |

## 5. Discussion

We applied the approach laid out by Kaewprasert et al. [1] who generated CIs for the mean and the difference between the means of several ZIG distributions by using the fiducial GCI and Bayesian and HPD interval methods. The optimal approach was discovered to be the HPD interval based on the Jeffreys rule prior. In addition, by utilizing fiducial GCI, we expanded Zhang et al. [14] method for constructing simultaneous CIs for distributions containing some zero observations. In the present study, we used the fiducial GCI, Bayesian, HPD interval, and MOVER approaches to construct CIs to compare the means of multiple ZIG distributions via simulation studies and using real rainfall datasets containing zero observations from six regions in Thailand.

The outcomes of the simulation study with a range of sample sizes and probabilities for nonzero values shed light on the analytical conduct of the simultaneous CIs. For $k = 3$ or 6, we discovered that the HPD interval based on the Jeffreys rule prior is the most suitable approach for all of the scenarios tested. The coverage probabilities and expected lengths of the 95% simultaneous CIs for $k = 3$ were comparable to those for $k = 6$ for various sample sizes. Moreover, the expected lengths of the approaches decreased as the probability of nonzero values was increased.

Importantly, the practicability of these methods was demonstrated by estimating the ratios of the means of multiple daily rainfall datasets in September 2021 for the six areas in Thailand. The selected rainfall station for each location had the same average number of rainy days, resulting in the probabilities of nonzero values being roughly the same. The results of this empirical application were in agreement with those of the simulation study results in that the HPD interval based on the Jeffreys rule prior was the most appropriate. Hence, it is possible to predict the ratio of rainfall in September of the following year in regions of Thailand that have an average chance of frequent rainfall. Therefore, our approach could be used to create an imminent natural alarm for natural disasters such as floods and landslides to alert people to make preparations in advance.

## 6. Conclusions

Herein, six methods for constructing simultaneous CIs for the ratios of the means of multiple ZIG distributions based on the fiducial GCI approach, Bayesian, and HPD interval approaches based on the Jeffreys rule or uniform prior and MOVER are presented. Their coverage probabilities and expected lengths from a simulation study indicate that the HPD interval based on the Jeffreys rule prior performed the best in most cases, while in some situations, the fiducial GCI performed well for both $k = 3$ and 6. Applying the methods to compare the rainfall datasets for September 2021 from six regions in Thailand shows that the HPD interval based on the Jeffreys rule prior and the fiducial GCI once again performed the best, which is consistent with the simulation results. Hence, constructing simultaneous CIs for the ratios of the means of multiple ZIG datasets should be carried out by using the HPD interval based on the Jeffreys rule prior. For some applications, we offer the fiducial GCI as an alternative approach. Researchers that are interested in analyzing rainfall means can use the R package we developed. Future studies will investigate into other statistical parameters like the coefficient of variation because they are important when making statistical inferences. In addition, we discovered that the coefficient of variation is an useful tool for evaluating rainfall dispersion. On CIs for the coefficient of variation of a zero-inflated gamma population, there are few research studies published. Therefore, we will investigate into this soon.

**Author Contributions:** Conceptualization, S.-A.N.; methodology, S.-A.N. and S.N.; software, T.K.; validation, T.K., S.-A.N. and S.N.; formal analysis, T.K. and S.-A.N.; investigation, S.-A.N. and S.N.; resources, S.N.; data curation, T.K.; writing—original draft preparation, T.K.; writing—review and editing, S.-A.N. and S.N.; visualization, S.N.; supervision, S.-A.N. and S.N.; project administration, S.N.; funding acquisition, S.-A.N. All authors have read and agreed to the published version of the manuscript.

**Funding:** This research has received funding support from the National Science, Research and Innovation Fund (NSRF), and King Mongkut's University of Technology North Bangkok: KMUTNB-FF-65-22.

**Institutional Review Board Statement:** Not applicable.

**Informed Consent Statement:** Not applicable.

**Data Availability Statement:** The real datasets of rainfall were obtained from the Royal Irrigation Department [22–27].

**Acknowledgments:** The first author wishes to express gratitude for financial support provided by the Thailand Science Achievement Scholarship (SAST).

**Conflicts of Interest:** The authors declare no conflict of interest.

## Abbreviations

The following abbreviations are used in this manuscript:

| | |
|---|---|
| AIC | Akaike information criterion |
| Baye.Jef | The Bayesian confidence interval based on Jefreys'rule prior |
| Baye.Uni | The Bayesian confidence interval based on uniform prior |
| CI | Confidence interval |
| CP | Coverage probability |
| EL | Expected length |
| GCI | Generalized confidence interval |
| GPQ | Generalized pivotal quantity |
| HPD | Highest posterior density |
| HPD.Jef | Highest posterior density based on Jefreys'rule prior |
| HPD.Uni | Highest posterior density based on uniform prior |
| MOVER | Method of variance estimates recovery |
| PB | Parametric bootstrap |
| ZIG | Zero-inflated gamma |

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
