# Peer review of "Simultaneous Confidence Intervals for the Ratios of the Means of Zero-Inflated Gamma Distributions and Its Application"

_mathematics, doi:10.3390/math10244724_

Round 1

Reviewer 1 Report

 In this paper authors have thoroughly studied simultaneous confidence intervals for the ratios of the means and its application in Zero-Inflated Gamma distributions. They applied the approach laid out by Kaewprasert et al. (2022) who generated CIs for the mean and the difference between the means of ZIG distributions by using the fiducial GCI and Bayesian and HPD interval methods. Therefore, they did not present a new idea in this manuscript, but it is well organized. Although the work motivation is not very high, since the estimation of the ratio of means is not much different from the difference of the means.

I provide following major remarks on this manuscript.

1-    Equation (1) should be revised since the value of delta is not considered.

2-    Correct typographical errors of mathematical relation in line 67.

3-    In order to abbreviate the terms, it must be done after they first appear. (Referring to line 77)

4-    In Table 4, for persuasion, more distributions should be considered in the goodness-of-fit test.

5-    The proof of relations of Section 2.2 is given in the appendix.

6-    In Section 4, a more complete explanation will be given about the tables and figures.

7-    Mathematical relationship used and assumptions in data simulation should be explained.

Author Response

 Dear Reviewers,
We are grateful for the reviewer’s valuable comments and have all suggestions
seriously. Reviewer’s critiques addressed section by section in this document, and
corrections were incorporated in manuscript accordingly
. Please see attached.

Reviewer 2 Report

see the attachment

Author Response

 Dear Reviewers,
We are grateful for the reviewer’s valuable comments and have all suggestions
seriously. Reviewer’s critiques addressed section by section in this document, and
corrections were incorporated in manuscript accordingly. Please see attached.

Reviewer 3 Report

I have read this article with great interest.

From my point of view this paper is a good work, however, I kindly ask the authors to consider the following points:

1) Do not use words that are shown in the title and/or in the abstract as keywords.

2) Please use a table for all abbreviations/acronyms.

3) Correct typos. There are some throughout the paper.

4) I think the introduction is a bit short. Improve it. I recommend citing more works.

5) If authors assign a number to an equation, then the authors must cite that equation. Remove the numbering of equations that are not cited.

6) I don't like the presentation of some equations. Check them all please.

7) Use a LaTeX package and improve the presentation of algorithms 1, 2, 3, and 4. Define input and output parameters for the algorithms. For example, the number of times steps 2 and 3 are repeated (2000 times) is also an input parameter $n$. Simply $n=2000$.

8) This is not mandatory: If the authors analyze well, the four algorithms can be unified in a single algorithm.

9) For the reader, the simulation study is a bit confusing. Please draw a diagram that graphically illustrates the simulation study carried out.

10) This is not mandatory: I have always believed that, when running simulations, reporting processing times is important. Also include the hardware used (processor, RAM, operating system).

11) Show, in a summarized way (for example in a table), the important conclusions that were obtained as a result of the simulation study. I know that this is already in the text, but it is essential to summarize it for the benefit of the reader.

12) There are parts of the document that do not have a good writing. Read the paper carefully and improve your writing.

13) I believe that both the abstract and the conclusions deserve another presentation. Please improve it.

14) What next? this is a good work. Have the authors considered creating an R package with their proposal so that it can be used by researchers? Be creative. What will you do next? Mention future work and possible applications.

I think that if the authors work carefully on the aforementioned points, I can recommend the publication of this nice paper.

Author Response

(The authors gave the same response as above.)

Author Response

(The authors gave the same response as above.)

Round 2

Reviewer 1 Report

Dear editor
The authors made the desired corrections and the revised manuscript is recommended for publication.

Reviewer 3 Report

The paper has been improved. I recommend the paper for publication.